# The In Vitro Assessment of Antibacterial and Antioxidant Efficacy in *Rosa damascena* and *Hypericum perforatum* Extracts against Pathogenic Strains in the Interplay of Dental Caries, Oral Health, and Food Microbiota

**DOI:** 10.3390/microorganisms12010060

**Published:** 2023-12-28

**Authors:** Maria Antoniadou, Georgios Rozos, Natalia Vaiou, Konstantinos Zaralis, Caglar Ersanli, Athanasios Alexopoulos, Athina Tzora, Theodoros Varzakas, Chrysoula (Chrysa) Voidarou

**Affiliations:** 1Department of Dentistry, School of Health Sciences, National and Kapodistrian University of Athens, 15784 Athens, Greece; mantonia@dent.uoa.gr; 2CSAP, Executive Mastering Program in Systemic Management, University of Piraeus, 18534 Piraeus, Greece; 3Department of Agriculture, School of Agricultural Sciences, University of Western Macedonia, 53100 Florina, Greece; clevervet@hotmail.com (G.R.); kzaralis@uowm.gr (K.Z.); 4Department of Agriculture, School of Agriculture, University of Ioannina, 47100 Arta, Greece; c.ersanli@uoi.gr (C.E.); tzora@uoi.gr (A.T.); 5Laboratory of Microbiology, Department of Medicine, National and Kapodistrian University of Athens, 11527 Athens, Greece; nvaou@hotmail.com; 6Laboratory of Microbiology, Biotechnology & Hygiene, Department of Agricultural Development, Democritus University of Thrace, 68200 Orestiada, Greece; alexopo@agro.duth.gr; 7Department Food Science and Technology, University of the Peloponnese, 24100 Kalamata, Greece

**Keywords:** plants extracts, food extracts, antibacterial efficacy, antioxidant effect, phenolic compounds, time-kill kinetics, minimal inhibitory concentration, antibiofilm activity, dental caries, oral health, holistic therapeutic approach, natural products chemistry, *Lactobacillus acidophilus*, periodontitis, *Streptococcus mutans*

## Abstract

The rising demand for novel antibiotic agents prompts an investigation into natural resources, notably plant-derived compounds. In this study, various extracts (aqueous, ethanolic, aqueous-ethanolic, and enzymatic) of *Rosa damascena* and *Hypericum perforatum* were systematically evaluated against bacterial strains isolated from dental lesions (*n* = 6) and food sources (raw milk and broiler carcass, *n* = 2). Minimal inhibitory concentration (MIC), minimal bactericidal concentration (MBC), antibiofilm activity, and time-kill kinetics were assessed across a range of extract concentrations, revealing a dose-responsive effect. Notably, some extracts exhibited superior antibacterial efficacy compared to standard clinical antibiotics, and the time-kill kinetics demonstrated a rapid elimination of bacterial loads within 24 h. The susceptibility pattern proved strain-specific, contingent upon the extract type, yet all tested pathogens exhibited sensitivity. The identified extracts, rich in phenolic and polyphenolic compounds, as well as other antioxidant properties, contributed to their remarkable antibiotic effects. This comprehensive investigation not only highlights the potential of *Rosa damascena* and *Hypericum perforatum* extracts as potent antibacterial agents against diverse bacterial strains including caries pathogens, but also underscores their rapid action and dose-dependent efficacy. The findings suggest a promising avenue for harnessing plant-derived compounds in the development of novel antimicrobial strategies against dental caries and other oral inflammations, bridging the gap between natural resources and antibiotic discovery.

## 1. Introduction

Natural medicines for therapeutic and preventive dental care and anticariogenic performance are an ancient cross-cultural practice [1,2]. Dental caries and periodontal diseases have long been the cause of missing and broken teeth as well as the severe loss of dental structures [3,4]. Ancient treatment methods for caries and other oral pathologies included the use of medicinal plants and animal products to maintain dental health and enhance the healing process [5,6]. Early healers also administered psychoactive herbal products to reduce stress and tooth pain in their patients [7,8].

Dental caries, a longstanding and widespread ailment, encompass both the disease itself and the resultant lesions arising from pH fluctuations [9]. The condition emerges when the oral biofilm microbiota, typically in a state of homeostasis, transitions to an acidogenic, aciduric, and cariogenic population, primarily due to consistent exposure to fermentable carbohydrates (glucose, fructose, maltose, and sucrose) in the diet [10]. While the initial consequences of this shift may be clinically imperceptible, over time, it can lead to discernible mineral loss within the tooth’s hard structures, culminating in a visible carious lesion. This dietary-microbial disease necessitates the formation of a cariogenic biofilm by tooth-adherent cariogenic bacteria, notably *Streptococcus mutans*, which metabolize sugars to produce acid [9]. Furthermore, periodontal disease is characterized by the progressive degradation of soft and hard tissues within the periodontal complex, driven by an intricate interplay between dysbiotic microbial communities and abnormal immune responses in gingival and periodontal tissues [11]. The enrichment of periodontal pathogens occurs as the resident oral microbiota undergo dysbiosis, triggering inflammatory responses that result in tissue destruction—a cyclical process involving proteolysis, inflammation, and the proliferation of periodontal pathogens. In the pertinent literature, periodontitis is associated with a decrease in the abundance of Proteobacteria and Actinobacteria, coupled with an increased prevalence of Bacteroidetes and Firmicutes [12]. These insights underscore the multifaceted nature of dental caries and periodontal diseases, emphasizing the critical role played by microbial dynamics and immune responses in the pathogenesis of these conditions. Contemporary minimally invasive dentistry prioritizes a preventive approach over traditional restorative measures for hard dental tissues and therapeutic interventions for soft tissues. To achieve this, various topical formulations such as gels, varnishes, mouthwashes, and dentifrices containing fluoride or chlorhexidine have been employed. While fluoride is a validated agent for caries prophylaxis, its excessive use can lead to fluorosis and cartilage hardening [13]. In the context of oral diseases, the widespread use of bactericides or antibacterial agents presents several drawbacks, including adverse effects on the gastrointestinal system and an elevated risk of resistance development to these chemicals [14]. Additionally, these agents may induce side effects such as tooth staining, altered taste sensation, toxic effects on connective tissues, dryness, soreness of the oral cavity, and oral desquamation [15]. Balancing the benefits of preventive interventions with an awareness of potential side effects is crucial in the pursuit of effective and patient-friendly oral healthcare strategies. To overcome these problems, instead of using the fluoride and other chemical substances that have been used so far for prevention and treatment in both caries and periodontal diseases, it has been proposed that medicinal plant extracts which influence the causative bacteria of tooth decay [16,17]] and periodontitis [18] could be used. For this purpose, efficacy has already been demonstrated for various herbal leaf extracts such as Tulsi, Neem, Guava, *Aloe vera*, Pudina, green tea, and Oolong tea [19]. Also, the extracts of *Azadirachta indica*, *Ocimum sanctum*, *Murraya koenigii* L., *Acacia nilotica*, *Eucalyptus camaldulensis*, *Hibiscus sabdariffa*, *Mangifera indica*, *Psidium guajava*, *Rosa indica*, and *Aloe barbadensis* have all been found to inhibit certain dental caries and periodontal pathogens [20,21]. Furthermore, the crude methanolic extract of rosemary is known to inhibit the growth of cariogenic streptococci while its activity against periodontal pathogens such as *Porphyromonas gingivalis* and *Prevotella intermedia* needs to be further evaluated [21]. The antimicrobial potential of *Rosa damascene* and *Rosmarinus officinalis* against *Streptococcus mutans* and *Streptococcus sanguinis* (*formerly S. sanguis*) has also been assessed, with positive results [22]. *Hypericum perforatum* L., commonly known as St. John’s wort, exhibits notable antioxidant and anticancer activities, demonstrating antiproliferative and cytotoxic effects against various cancer cells [20]. Additionally, it shows potential in addressing oral ulcers, bacterial infections, and oral pain control [23,24]. A holistic, natural approach to oral disease control includes the consumption of fermentative foods [25], probiotics, prebiotics, synbiotics [26], and nutritional beverages such as olive oil [27] or honey [28,29]. Ginger rhizome (*Zingiber officinale* Roscoe, Zingiberaceae) is recognized in the literature for its proven antimicrobial activities among natural food sources [30]. Furthermore, unfermented cocoa, red grape seed, and green tea have demonstrated efficacy in inhibiting plaque bacteria, glucosyltransferase activity, glucan, and plaque formation [31,32,33]. Notably, extracts from these sources are non-toxic and approved by the United States Food and Drug Administration (FDA) [34]. These findings underscore the potential of diverse natural sources in providing safe and effective means for a comprehensive, natural approach to oral health encompassing both preventive and therapeutic measures and recognizing nature’s potential in disease prevention [35].

In response to the economic considerations driving the demand for cost-effective therapies for oral diseases, research has intensified, acknowledging the persistent global burden of oral health issues, particularly dental caries [36]. Dental caries remains a significant public health concern, affecting nearly 3.5 billion people globally, with a substantial impact on both permanent and primary teeth [37]. The consequences of dental diseases extend beyond discomfort, affecting aesthetics and function, and are associated with systemic health problems [38,39]. Given the prohibitive costs of treating established dental diseases, especially for underserved populations, there is a growing demand for safer and more effective medications with dental applications, leading to increased research on medicinal plants and food [36,40,41]. This aligns with the broader paradigm shift in healthcare towards comprehensive plans that integrate oral health into holistic well-being, emphasizing lifestyle and nutritional changes to foster both oral and general health. The challenges faced by the dental healthcare sector, such as inequality in access and affordability concerns, necessitate strategic interventions for underserved populations, including mobile dental units and community health initiatives [37]. Additionally, ongoing education and awareness initiatives are crucial to dispel cultural misconceptions and disseminate accurate information on oral health practices [38]. Sustainable solutions in natural products inhibiting cariogenic biofilms highlight the potential of plant-based therapies in preventing dental caries while addressing sustainability issues in oral healthcare [36]. This approach not only offers effective prevention and therapy for dental caries but also contributes to a more sustainable and environmentally conscious approach to dental care, aligning with global efforts for improved oral health outcomes [35].

Under this scope, the present study aims to contribute updated insights into the antimicrobial effects of plant extracts on caries and periodontal diseases. Specifically, various extracts (aqueous, ethanolic, aqueous–ethanolic, and enzymatic) of *Rosa damascena* and *Hypericum perforatum* are comprehensively tested against strains isolated from dental lesions and food sources (raw milk and broiler carcass), an aspect not previously addressed collectively. Recognizing food as a potential vehicle for pathogens entering the oral cavity, this research employs modern methods, including MIC (minimal inhibitory concentration), MBC (minimal bactericidal concentration), antibiofilm activity, and time-kill kinetics. By scrutinizing the antimicrobial properties of these extracts on microorganisms from real-world oral and food environments, the article aims to augment contemporary literature in the field.

## 2. Materials and Methods

### 2.1. Plant Materials

Two local herbs were used as plant material in our study. The herbs were collected from different locations in Greece: (a) roses (*Rosa damascene*) were collected from local producers in Kozani, an area of Western Macedonia and (b) St. John’s wort (*Hypericum perforatum*) was harvested from an area of Epirus. The collected plant samples were left to dry at room temperature. Once completely dried, samples were separated into different plant parts: flowers, roots, leaves, bark, and stems. Dried plant materials (for the Roses, only rose petals, calyces, and pollen and for *H. perforatum,* only leaves and flowers) were then crushed to powder using a high-speed grinder in order to obtain a manageable material and facilitate the extraction procedure. The raw material was stored until the beginning of the study at −18 °C.

#### 2.1.1. Extraction Method for Plants

The maceration method was used to prepare the aqueous, ethanolic extract and enzymatic extract.

#### 2.1.2. Aqueous Extract

The aqueous extract of roses used in the present study came from the Women’s Cooperative of Kozani. The preparation technique is as follows. Approximately 10 g of powdered plant were dissolved in 90 mL of boiled distilled water to make 100 mL of aqueous extract (10% *w*/*v*). The mixture was placed at room temperature for 24 h in sterile flasks with constant stirring while ensuring the nozzle was closed and then was filtered through sterilized Whatman No. 1 filter paper. After filtration, the extracts were concentrated by evaporation under reduced pressure using a rotary evaporator (KNFRC 900, KNF Neuberger GmbH, Breisgau, Germany). For *H. perforatum*, aqueous extracts were prepared by immersing plant powder material in sterile distilled water. Plant powder was suspended in distilled water (dw) at a ratio of plant powder: dw of 1:2 (*w*/*v*), and then placed in the platform shaker incubator at 30 °C for 24 h. For both plants, the liquid extracts were stored at −80 °C for 24 h before lyophilization. Aqueous extracts were lyophilized using a freeze dryer for approximately 7 h or overnight. Before use, aqueous extracts were placed under ultraviolet light overnight to eliminate possible microbial contaminants.

#### 2.1.3. Ethanolic Extract (E)

To obtain ethanol extracts we performed the same procedure mentioned in Section 2.1.2 for the *Hypericum perforatum*, where the solvent ethyl alcohol solution was used instead of distilled water. Two concentrations of 96% ethyl alcohol solution were used, 40% and 60% (prepared from distilled water).

#### 2.1.4. Enzymatic Extract (ENZ)

A total of 1 kg of pretreated fresh plant (for *R. damascene* this included rose petals, calyces, and pollen and for *H. perforatum* this included leaves and flowers) was immersed in a solution of 2 kg of dw acidified to pH = 2 using concentrated hydrochloric acid and 1.0% pepsin (Merck KGaA, Darmstadt, Germany) [42]. Briefly, the pretreated procedure includes washing under running water, manual dressing to remove unsuitable elements for further processing and cutting the parts into 1 mm pieces using a slicer to increase the surface area. Then, the pieces were washed with PBS to remove intracellular vehicles released from broken cells. After incubation at 37 °C for 48 h, hydrolysis was interrupted by heating for 10 min. The obtained solution was divided into smaller batches of 200 g each, which were squeezed manually with the help of a sterile pestle, and each batch was filtered through sterilized Whatman No.1 filter paper. Finally, the solvent of each batch was evaporated by means of a rotary evaporator (KNFRC 900, KNF Neuberger GmbH, Breisgau, Germany). Deep freezing at −80 °C and subsequent lyophilization followed as the last step.

Acronym names for the obtained extracts was used: A—aqueous extract, E40—ethanolic extract (40% *v*/*v* aqueous ethanol), E60—ethanolic extract (60% *v*/*v* aqueous ethanol), and ENZ—enzymatic extract. Samples were prepared by weighing out the above-mentioned crude extracts and calculating the volume of solvent [5% aqueous solution of dimethyl sulfoxide (DMSO)] to be added to create a sample stock solution of 100 mg/mL concentration. Sterile distilled water was used to dissolve aqueous extracts. All of the following experiments were conducted in triplicate.

### 2.2. The Quantification of the Biological and Antioxidant Activities of Plant Extracts

To evaluate the phytochemicals contained in the plant extracts, the following tests were used [43]. *Clarification*: The aqueous extract dissolved in sterile boiled water was marked as A, while the one dissolved by a solvent of a 1/1 ratio of boiled water and 95% methanol respectively, was marked as A*.

#### 2.2.1. The Detection of Alkaloids

A quantity of 50 mg of solvent-free extract was placed and mixed in a test tube with 2 mL of diluted hydrochloric acid and filtered, then the following tests were performed: (a) a few drops (2–3) of Mayer’s reagent were added, where a white/creamy or yellow precipitate indicated the presence of alkaloids; (b) the test with Wagner’s reagent where a reddish-brown precipitate confirmed the presence of alkaloids; and (c) the test with Hager’s reagent (Picric acid test) where the appearance of a prominent yellow precipitate showed the test to be positive [43,44].

#### 2.2.2. The Detection of Anthraquinones

Fifty milligrams of the extract were dissolved in 5 mL dw and then filtered. Two milliliters of the solution of the extract were put in a test tube with 10 mL of benzene and the mixture was shaken vigorously for 10 min and filtered. Finally, 5 mL of 10% ammonia solution were added to the test tube and again shaken vigorously for 30 s. By a rating of positivity, a pink, red, or violet color indicated the presence of free anthraquinones [43,44].

#### 2.2.3. The Detection of Terpenoids (Salkowski Test)

A quantity of 200 mg of solvent-free extract was put into a test tube and mixed with 2 mL of chloroform and filtered. Then, concentrated sulfuric acid (H_2_SO_4_) (a few drops) was added to form a layer and gentle agitation followed. Then, the mixture was allowed to stand. A reddish-brown coloration at the interface confirmed the presence of tri-terpenoids [44].

#### 2.2.4. The Detection of Saponins (Foam Test)

A total of 500 milligrams of the extract were dissolved in 2 mL of dw. The suspension was shaken for 15 min in a graduated cylinder. The presence of saponins was established by the formation of a 2 cm layer of foam [43].

#### 2.2.5. The Detection of Tannins (Ferric Chloride Test—Braymer’s Test)

A total of 500 milligrams of the extract were dissolved in 5 mL of dw and boiled for 3 min. Then the solution was clarified by filtration and transferred to a new test tube which contained 3 mL of dw and a few drops of 10% ferric chloride (FeCl_3_) were added. The presence of tannins was established by the development of a dark green color [44].

#### 2.2.6. The Detection of Cardiac Glycosides (Keller–Kiliani Test)

Fifty milligrams of the extract were dissolved in dw and then filtered. To a quantity of 2 mL of filtrate, 1.5 mL of glacial acetic acid, a drop of 5% ferric chloride (FeCl_3_), and a drop of concentrated sulfuric acid H_2_SO_4_ (along the side of walls of the test tube) were added. The change of green blue at the upper layer and reddish brown at the junction of two layers confirmed the presence of cardiac glycosides [43,44].

#### 2.2.7. Total Phenolic Concentrations

The concentration of total phenolic compounds in the plant extracts was determined colorimetrically by the Folin–Ciocalteu method, using gallic acid as a standard. The total concentration of phenolic compounds was then expressed as gallic acid equivalents (GAE) in milligrams per g of sample [45,46].

#### 2.2.8. The Estimation of Total Flavonoid Content

Total flavonoid content was determined by the aluminum chloride assay according to Park et al., 2008 [33,47]. The total flavonoids were expressed as milligrams of catechin equivalents (CE) per g of sample.

#### 2.2.9. Total Antioxidant Activity Assay (DPPH Free Radical Scavenging Assay)

The free radical scavenging activity of the fractions was evaluated in vitro using the *2,2-diphenyl-1-picrylhydrazyl* (DPPH) assay. The stock solution was prepared by diluting 24 mg DPPH in 100 mL methanol and the dilution was stored at 20 °C. The working solution was obtained by further diluting the DPPH solution with methanol until an absorbance of about 0.98 ± 0.02 at 517 nm was attained using the spectrophotometer. A 3 mL aliquot of this solution was mixed with 100 μL of the sample at various concentrations (1–500 μg/mL). The reaction mixture was shaken well and incubated in the dark for 15 min at room temperature. Then the absorbance was measured at 517 nm. [47,48]. The control was prepared as above, consisting only of the solvent. The scavenging activity was calculated based on the percentage of DPPH radical scavenged using the following equation:Scavenging effect % = [control absorbance − sample absorbance/control absorbance] × 100

#### 2.2.10. Reducing Power Assay

The reducing power was based on the reduction of Fe (III) to Fe (II) in the presence of the solvent fractions and was estimated using the method described by Saeed et al., 2012 [47].

### 2.3. The Determination of the Antibacterial Activity In Vitro

#### 2.3.1. Tested Microbial Strains and Antibiotic Sensitivity Pattern

The strains of the pathogenic bacteria that were tested as cell targets, were as follows:-*Staphylococcus aureus* subsp. *aureus*, methicillin, and vancomycin resistant (source: dental caries area);-Methicillin-Resistant *S. aureus*, from raw milk;-Methicillin-Resistant *S. aureus*, from raw poultry;-*Streptococcus mutans* (source: oral cavity);-*Streptococcus salivarius* (source: oral cavity);-*Fusobacterium nucleatum* (source: oral cavity);-*Porphyromonas gingivalis* (source: oral cavity);-*Prevotella intermedia* (source: oral cavity);-*Parvimonas micra* (source: oral cavity);-*Staphylococcus aureus* subsp. *aureus*, reference strain ATCC 6538.

All of the above strains were identified and classified using standard laboratory procedures, which are followed by the National and Kapodistrian University of Athens, Departments of Medicine and Dentistry.

*Clarifications on nutrient substrates and bacterial growth conditions*: Since in the present study three Gram^+^ isolates and one reference strain (belonging to species *S. aureus*); two facultative anaerobic Gram^+^ isolates, which included *S. mutans* and *S. salivarius;* and four obligate anaerobic Gram^−^ isolates, *F. nucleatum*, *P. gingivalis*, *P. intermedia,* and *P. micra* were tested, the use of different media and incubation–growth conditions are required.

For *Staphylococcus aureus* isolates, including the reference strain, the procedures described in the following subsections were carried out using blood agar media for the overnight growth at 37 °C, under aerobic conditions, sterile saline solution (bioMérieux, Marcy-l’Étoile, France) to adjust the inoculum of fresh colonies to the McFarland unit at 0.5 (~1 × 10^8^ CFU/mL), and Mueller–Hinton agar (Oxoid, Hampshire, UK) for the determination of the antimicrobial activities of extracts.

For *S. mutans* and *S. salivarius* isolates, the main cultivation medium was Brain Heart Infusion (BHI, Difco Laboratories, Detroit, MI, USA) agar asnd broth, and the same value of the McFarland unit was kept, as above (0.5), but followed by incubation under anaerobic conditions at 37 °C using anaerobic rectangular jars with anaerocult A gas packs (Merck, Darmstadt, Germany).

For the obligate anaerobic Gram^−^ tested bacterial strains the recultivation process was carried out in Brain Heart Infusion (sBHI; Oxoid, Hampshire, UK) media supplemented with 5 μg/mL hemin and 1 μg/mL menadione [49,50,51]. Subsequently, colonies were suspended in sBHI broth, which was boiled for 20 min before use to reduce the oxygen content, to achieve a density corresponding to 1.0 McFarland turbidity standard. Finally, the main nutrient media for the investigation of the antimicrobial properties of the tested plant extracts was 5% defibrinated sheep blood Brucella agar (Merck, Germany) enriched with 5 μg/mL hemin, 1 μg/mL menadione, and 2 g/L yeast, under strictly anaerobic incubation conditions (using anaerobic rectangular jars with anaerocult A gas packs), taking care to avoid an excess of moisture by the addition of 4–5 drops of glycerol onto a piece of filter paper in an uncovered petri dish along with the plates in the jar.

#### 2.3.2. Antibiotic Susceptibility Assay

The antibiotic sensitivity of the used bacterial strains, and of the reference strain as well, was detected using the Kirby Bauer’s disk diffusion method, according to the standards set by The National Committee for Clinical Laboratory Standards (later renamed The Clinical Laboratory Standard Institute-CLSI). Based on the relevant clinical guidelines [51,52,53,54], the following antibiotics were included in the analyses of antimicrobial profiles: β-lactams [(amoxicillin/clavulanic acid: AMC, 20/10 µg); aminopenicillins (amoxicillin: A, 30 µg); markers for methicillin resistance (oxacillin, flucloxacillin)], especially for the strains of *Staphylococcus aureus*; Glycopeptide (Vancomycin; VA, 30 µg); second-generation cephalosporins (cefuroxime; CFX, 30 μg); third-generation cephalosporins (cefotaxime; CFT 30 μg); clindamycin (CLI, 2 μg); aminoglycosides (Gentamicin; GEN, 10 μg); macrolides (erythromycin; ERY, 15 μg); tetracycline (TER, 30 μg), fluoroquinolones (ciprofloxacin; CIP, 5 μg); carbapenems (Imipenem; IMI, 10 μg); and nitroimidazole (Metronidazole; MET, 5 µg). Also, the minimum inhibitory concentration (MIC) values of the above-mentioned antimicrobial agents against the tested bacterial strains were measured using the standard reference methods of the broth microdilution method according to the EUCAST and CLSI requirements [55,56,57,58] of each EO and the reference antimicrobials.

#### 2.3.3. Diffusion Test in Agar

As a first step in order to evaluate the antibacterial activity of the studied extracts, the disk diffusion method was used [54]. All of the details mentioned in Section 2.3.1 regarding the nutrient’s media used and the incubation conditions were followed. Briefly, for each tested bacterial strain a prepared inoculum from an overnight culture was immediately spread out on dried Mueller–Hinton agar, Brain Heart Infusion agar, and Brucella Agar with 5% sheep blood with hemin and menadione plates, for *S. aureus*, *Streptococcus,* and obligate anaerobic Gram^−^ isolates, respectively. Sterile 6 mm diameter Whatman paper No. 1 disks were impregnated with 10, 20, 50, and 100% (*v*/*v*) of each herb extract diluted in a 5% (*v*/*v*) aqueous solution of dimethyl sulfoxide, DMSO (Honeywell, Charlotte, NC, USA), or sterile distilled water, for aqueous extracts. A distilled water-loaded disk, also, was used as a negative control. Then, the plates were maintained at room temperature for 2 h to allow the tested bacteria to diffuse in the media agar (always taking care not to disturb the anaerobic conditions for the obligate anaerobic bacterial strains). Subsequently, the plates were incubated at 37 °C overnight for *S. aureus* stains, for strictly 36 h for *S. mutans* and *S. salivarius* tested strains and for at least 72 h for the obligate anaerobic Gram^−^ strains, for which anaerobic conditions were maintained throughout the incubation period, using anaerobic rectangular jars with anaerocult A gas packs. The results were observed at the end of the required incubation time and the zone of inhibition level was measured. Double triplicates were maintained for all of the experiments.

#### 2.3.4. Minimum Inhibitory Concentration and Minimum Bactericidal Concentration Determination

For an accurate estimation, and in much lower concentrations than in agar diffusion experiments, the minimum inhibitory concentration was estimated by employing the microdilution broth method in 96-well microplates [55,59]. Dilutions of plant extracts were used: extracts were diluted using 400 mg/mL crude extraction/5% aqueous solution of dimethyl sulfoxide (DMSO, except in the case of aqueous extracts, for which an ultrapure water solution was used, vortexed at 1500 rpm for 3 min and filtered through a 0.45 Whatman TM syringe filter (Merck, Germany). From these solutions, serial dilutions, using an ultrapure water solution, were prepared at the following concentrations: 200 mg/mL, 100 mg/mL, 50 mg/mL, 25 mg/mL, 12.5 mg/mL, 6.25 mg/mL, 3.125 mg/mL, 1.56 mg/mL, 0.78 mg/mL, 0.39 mg/mL, 0.195 mg/mL, and 0.0975 mg/mL. The inoculum was prepared and regulated to standard 1 of the Mc Farland scale and all aerobic or facultative anaerobic strains were inoculated in double-strength Mueller–Hinton broth (MHB) and anaerobic bacteria were inoculated in sBHI broth, supplemented with 5 μg/mL hemin and 1 μg/mL menadione. The incubation times and conditions listed in Section 2.3.3 were followed. To verify the process, wells containing solely the liquid medium or the liquid medium with inoculum or chlorhexidine (CHX) 0.2% served as controls. After incubation, the liquid media located in each well was stained with an aqueous resazurin solution (Sigma-Aldrich, St Louis, MO, USA) at a concentration of 0.02%. and the 96-well microplates were re-incubated for another two hours. The lowest concentration corresponding to the test well that maintained the blue resazurin staining was interpreted as the MIC. A change in color to pink-purple or pink indicated that resazurin was reduced and marks the presence of bacterial growth. All assays were performed in triplicate.

The MBC (Minimum Bactericidal Concentration) examination involved the transfer of a 20-μL aliquot from each well where no growth was observed and from the well corresponding to the MIC reading onto the plates with Mueller–Hinton Agar, or, in the case of anaerobic bacterial strains 5%, defibrinated sheep blood Brucella agar enriched with 5 μg/mL hemin, 1 μg/mL menadione and 2 g/L yeast. The plates were incubated at 37 °C for 24 h, 36 h, and 37 h for *S. aureus*, facultative anaerobic *Streptococcus* strains, and obligate anaerobic Gram^−^ isolates, respectively. Finally, the growth of colonies on the plates was confirmed at the end of that period. MBC was defined as the lowest concentration of extract which resulted in a complete elimination of bacteria.

#### 2.3.5. The Evaluation of the Anti-Biofilm Properties of the Studied Herbal Extractions

Before determining the anti-biofilm effects of the tested plant extracts, the biofilm formation of the studied bacterial isolates was investigated [60]. To determine the formation of biofilm of the bacterial isolates, a semi-quantitative method of biofilm determination was performed in collagen type I-coated 96-well flat-bottom microplates (hermo Scientific™Nunc™, Waltham, MA, USA). Initially, a suspension of fresh bacterial culture in Trypticase Soy Broth (TSB, Sigma-Aldrich) medium supplemented with 1% (*w*/*v*) glucose (TSBG) for the tested aerobic or facultatively anaerobic strains and Tryptic Soy Broth supplemented with yeast extract (5 g/L), L-cysteine hydrochloride (0.5 g/L), hemin (5 mg/L), and menadione (1 mg/L) for obligate anaerobic Gram^−^ isolates [61] were prepared. Then, 100 mL of the bacterial suspension (for each tested bacterial isolate the turbidity was adjusted to 10^8^ CFU/mL) were added to the end of each well of the collagen type I-coated 96-well microplate and incubation at 37 °C under aerobic conditions (for 24–36 h), or anaerobically (for at least 48 h), followed. After the required incubation time, the wells were washed with phosphate buffered saline X1 (pH: 7.4) twice and fixed with 150 mL absolute methanol for 10 min. Finally, fixed bacterial cells were stained with 0.1% crystal violet dye (Sigma-Aldrich, Dorset, UK) for 30 min at room temperature. Then, the excess crystal violet was removed by washing, and the number of attached cells was measured by the process of re-solubilization, adding 160 mL of 33% (*v*/*v*) glacial acetic acid. The absorbance (OD) at 595 nm was measured. Wells containing TSB only served as a negative control or blank. Each test was repeated by three independent experiments for each of the tested isolates and their average OD was calculated at a wavelength of 595 nm. The value obtained was compared with the cut-off value (ODc). ODc is defined as three standard deviations above the mean OD of the negative control [62]. Based on the results, the isolates were classified as follows: non-biofilm producers (OD ≤ ODc); weak biofilm producers (ODc < OD ≤ 2 × ODc); moderate biofilm producers (2 × ODc < OD ≤ 4 × ODc); and strong biofilm producers (4 × ODc < OD).

To check the anti-biofilm properties of each studied herbal extraction, the same procedure mentioned above, in Section 2.3.5, was used, with the only difference being that the bacterial suspension inside of each well was co-cultured with a sub-MIC concentration of plants extract as treatment [63,64]. The control sample consisted of wells which did not receive any portion of the plant extracts.
$$Percentage\;\left(\%\right)\;inhibition=\frac{OD\;Negative\;control-OD\;Sample}{OD\;Negative\;control}$$

Based on the results, the tested herbal extracts were classified as: Excellent (++++) ABF activity (>95% inhibition); Very Good (+++) ABF activity (>95–80% inhibition); Good (++) ABF activity (>80–50% inhibition); Poor (+) ABF activity (more than 0–50% inhibition); No (−) ABF activity (0% or less).

#### 2.3.6. The Analysis of Time-Kill Kinetics

After obtaining the MIC of each herbal extraction using the micro-broth dilution method, its bactericidal effects were evaluated by using the Time-Kill Kinetics evaluation method. The tested inhibiting factors were the aqueous (A) and enzymatic extracts (ENZ) of roses. Initially, 1 × 10^6^ CFU/mL of the bacteria (1 × 10^8^ CFU/mL for the anaerobes) was treated with concentrations equal to the MIC, twice the MIC, four times the MIC, and eight times the MIC of the tested extracts, followed by incubation at 37 °C for 24 h, 36 h, and 48 h for *S. aureus*, facultative anaerobic *Streptococcus* strains, and obligate anaerobic Gram^−^ isolates, respectively. Then, at time intervals of 0, 2, 4, 8, 12, and 24 h, 0.1 mL of the treatment microbial suspension cultured in the broth was transferred to the surface of the TSA medium for the aerobic bacteria and 5% defibrinated sheep blood Brucella agar enriched with 5 μg/mL hemin, 1 μg/mL menadione, and 2 g/L yeast, for anaerobes. After 24–48 h, the number of colonies was counted. The culture medium containing the tested bacteria without the presence of an herbal substance was used as a control. Each procedure was performed in triplicate (three independent experiments) [65,66].

### 2.4. Statistical Analysis

All microbial counts were expressed as log CFU and presented as mean ± standard deviation. Normality was checked with the Shapiro–Wilk test. Comparisons of the means between various groups of data were performed by using the Kruskal–Wallis test or ANOVA (for normally distributed data) with Tukey’s HSD post hoc comparison. Correlations were estimated with the Spearman rank correlation coefficient. In all cases significance level was 95%. SPSS v28 (IBM Corp. Armonk, NY, USA) was used to perform the statistical estimations.

## 3. Results

### 3.1. Screening the Phytochemical Pattern—The Antioxidant Activity of Plant Extracts

Table 1 shows the groups of phytochemical compounds detected either in *R. damascene* or *H. perforatum* samples. Roughly, all *H. perforatum* extracts were richer in anthraquinones than *R. damascene*, while all groups of phytochemicals were detected in the enzymatic extracts of both species.

The total phenolic content significantly differs among the various extracts of *R. damascene* (Kruskal-Wallis’s test statistic: 13.03, *p* < 0.05) with the highest value observed in aqueous (104.92 ± 6.05 mg GAE/g of dried sample) and aqueous/methanolic fractions (89.1 ± 2.2) while the lowest value was observed in the 60% ethanolic sample (75.47 ± 1.79 mg GAE/g). Such differences were also observed in total flavonoid content in the same plant, with the highest values observed in the enzymatic (42.78 ± 0.41 mg CE/g) and aqueous extracts (41.97 ± 0.34 mg GAE/g), while the lowest values were observed in the 60% ethanolic extract (20.68 ± 0.5 mg CE/g). In *H. perforatum* extracts, the highest value of total phenolics was recorded in the 60% ethanolic extract (92.39 ± 2.06 mg GAE/g) and the lowest value was recorded in the enzymatic extract (49.2 ± 0.83 mg GAE/g). The highest value of total flavonoids was observed in enzymatic extract (33.63 ± 1.3 mg CE/g). Again, there were statistically significant differences regarding total phenolics or total flavonoids between the various *H. perforatum* extracts as indicated in Table 2, Figure 1.

The highest scavenging activity among the various *R. damascene* extracts occurred in Rosa A and A* at the highest concentration (500 and 300 μg/mL). In contrast, *H. perforatum* 40% and 60% ethanolic extracts were those with the highest antioxidant capacity in DPPH assays. As shown in Table 3, there were significant differences between DPPH values among the various extracts in specified concentrations.

Increased extracts concentrations resulted in higher reducing power in these samples (Table 4). Regarding *R. damascene*, the highest absorbance values were recorded in concentrations of 200 or 250 μg/mL of ethanolic extracts (40% and 60%) and of enzymatic extracts as well, reaching the highest absorbance value of 1.54 ± 0.01. In contrast, such high values (Abs 1.61 ± 0.01) were reached from the enzymatic extract of *H. perforatum* in an elevated concentration (250 μg/mL). The reducing power shows good linear relation in all sample extracts with R-squared values between 0.94 and 0.98. The linearity of the gallic acid standard was 0.69–0.70, in contrast to a linearity of 0.98–0.99 for the ascorbic acid.

### 3.2. The Analysis of the Antimicrobial Activity of Plant Extracts

The antibiotic susceptibility of the tested pathogens and the MIC of the antibiotics produced zones of inhibition from 9 mm (*P. intermedia*) to 36 or 37 mm (*S. salivarius*, *S. mutans*) with amoxicillin (30 μg); 24–35 mm with amoxicillin with clavulanic acid; 18 to 27 mm with vancomycin (30 μg); 10 to 35 mm with imipenem (10 μg); 0 to 26 mm with erythromycin (15 μg); 0 to 29 mm with clindamycin (2 μg); 0 to 266 mm with gentamycin (10 μg); 0 to 28 mm with tetracycline (30 μg); 0 to 52 mm with ciprofloxacin (5 μg); 21 to 27 mm with metronidazole (5 μg); 10 to 30 mm with cefuroxime (30 μg); and 8 to 31 mm with cefotaxime (30 μg) (Table 4).

All of the tested pathogenic bacteria were resistant to at least two of the various antibiotics, with *F. nucleatum*, *P. gingivalis*, *P. intermedia,* and *P. micra* demonstrating less resistance than the three *S. aureus* strains and the two Streptococci to amoxicillin (with and without clavulanic acid), vancomycin, imopenem, cefuroxime and cefotaxime. The 60% ethanolic and enzymatic extracts of *H. perforatum* with MIC values of 0.8 ± 0 mg/mL and 0.4 ± 0 mg/mL, respectively, were more effective against an *S. aureus* MSRA/VRSA strain isolated from dental septicemia than most of the commercial antibiotics. Only clindamycin and ciprofloxacin presented better inhibitory effects than the abovementioned *H. perforatum* extracts.

Oral pathogens, and especially those involved in periodontitis, are among the most resistant to the antibiotics. Various studies have indicated not only the increased percentage of resistant isolates [66,67,68,69], but also a trend over the years towards decreasing susceptibility profiles as in the study of Jepsen et al., (2021), with antibiotic non-susceptibilities observed in 37% of patients in 2008 and in 70% in 2015 [70].

#### The Disk Diffusion Assays of the Herb Extracts

The antibacterial activities of *H. perforatum* extracts from the disk diffusion experiments are presented in Table 5. Overall, all herb extracts in almost all concentrations (10, 20, 50 and 100%) were effective against pathogens and the reference strain. Concerning the strain *S. aureus* MRSA/VRSA, the *H. perforatum* enzymatic extract (100% concentration) had the most antimicrobial effect with a 48.2 ± 1.1 mm mean diameter zone of inhibition, followed by the E60 extract (31.7 ± 7.5 mm). Similarly, the enzymatic extract was equally effective against all other pathogens with inhibition zones ranging from 29.2 to 40.7 mm followed by the E60 extract with inhibition zones from 20 ± 0.2 to 43.1 ± 1.6 mm. Aquatic and E40 extracts were effective but only at higher concentrations (50–100%). In fact, when all data are considered, the aqueous and 40% ethanolic extracts presented similar mean zones of inhibition (17.4 to 17.5 mm) compared with the 60% ethanolic and enzymatic extracts (21.6 and 23.1 mm), comprising two discrete homogenous groups (Kruskal–Wallis *p* < 0.05). Additionally, positive and statistically significant correlations were recorded between the concentration of each extract and the zone of inhibition of each pathogen indicating a dose–response effect (Table 6).

The effectiveness of extracts from *R. damascene* against the pathogens varied considerably in terms of the type or concentration of the extract (Kruskal–Wallis *p* < 0.05). Most effective were the enzymatic and E60 extracts with respective mean inhibition zones of 16.65 ± 6.6 mm and 16.58 ± 5.0 mm. However, the aqueous extract was equally effective to the above against *S. mutans*, *P. gingivalis,* and *P. intermedia* (Table 7). A dose–response effectiveness was also revealed for the extracts of *R. damascene* (Table 8).

Minimum Inhibitory (MIC) and Bactericidal (MBC) Concentrations of extracts and tested antimicrobial agents.

Antibacterial activity, as indicated by the Minimum Inhibitory Concentration (MIC) and Minimum Bactericidal Concentration (MBC), against nine pathogens and a reference strain is presented in Table 9 and Table 10 for the two plant species, respectively. All extracts exhibited strong activities against pathogens, however, the strongest were observed when enzymatic extracts were used. In the case of *H. perforatum* experiments, this difference was more obvious in contrast to *R. damascene* results, where the strongest activity was also profound for the mixed ethanolic 60% (E60) extract.

The results from the disk diffusion experiments to assess the antibacterial activity of the various extracts against the nine pathogens and a reference strain are presented in Table 11. As shown, all *H. perforatum* extracts were more effective than those from *R. damascene,* with the enzymatic and the 60% ethanolic extracts producing the larger inhibition zones.

-Antibiofilm effectiveness and inhibition zone results

Since the antibiofilm activity was evaluated as a four-score variable regarding the inhibition of biofilm formation, i.e., poor (0–50% inhibition), good (50–80% inhibition), very good (80–95% inhibition), and excellent (>95% inhibition), the statistical analysis was based on medians or grouped medians instead of the means. In Figure 2, the median scores of the various herbal extracts are presented as well as the scores of the seven antibiotics. Overall, the antibiotics had a wide and strong antibiofilm effectiveness against all pathogens (where applied), with vancomycin being the most effective. However, the herbal extracts exhibited noticeable antibiofilm properties as derived from the inhibition zones which often were comparable to those of chemical drugs (Figure 3). Among the herbal extracts, *H. perforatum* ethanolic, mixed, and enzymatic extracts showed the highest values of biofilm inhibition, followed by the extracts of *R. damascene*.

In general, only the concentration of chemical antibiotics was correlated to antibiofilm effectiveness with Spearman correlation coefficients ranging from 0.64 to 0.81 (*p* < 0.01) (Table 12). Aqueous, ethanolic, enzymatic, or mixed extracts (i.e., aqueous/ethanolic) exhibited no such correlations when all data were considered. Individual positive correlations were observed for *H. perforatum* aqueous extract (r: 0.32, *p* < 0.05). However, although significant, such coefficients were well below the 0.7 to be considered strong enough. Such differences between chemical antibiotics and extracts are often attributed to the mixed nature of extracts as they contain compounds with different polarities and thus, different solubilities in aqueous test systems [71].

-Time of Kill Kinetics

Results showed that the most efficient concentrations for the strong reduction of the initial population of the pathogens were two times the MBC of each extract and above (×4 and ×8). Among the most resistant strains were *S. aureus* MRSA/VRSA, *S. aureus* MRSA (raw milk), and *S. aureus* MRSA (raw poultry), since the highest concentrations of *R. damascene* extracts required (8× MBC) and 24 h contact time to kill 99.99% of the exposed cells. The *S. aureus* reference strain was also resistant, although in this case two times the MBC and 24 h contact time were needed to eliminate almost all cells. The rest of the pathogens were killed after 8 h (8× MBC), or after 24 h when extracts at concentrations 2× MBC or 4× MBC were used (Table 13; Figure 4). In all cases the statistical difference from the control was significant. The enzymatic extract was more effective than the aqueous extract in the case of *S. aureus* MRSA/VRSA, S. aureus MRSA (raw milk), and *S. aureus* MRSA (raw poultry) since in most of the cases, killing was achieved in half of the time (Table 14; Figure 4). However, this wasn’t the case with the other pathogens, where the aqueous *R. damascene* extract was equally, or even the most, efficient.

## 4. Discussion

In the face of diminishing efficacy resulting from the widespread resistance to conventional antibacterial medications employed in clinical settings, there is a pressing need to channel research efforts toward the creation of innovative compounds endowed with antibiotic properties. The vast and diverse spectrum of nature provides a compelling realm for exploration, offering a trove of natural substances that warrant investigation for their potential therapeutic attributes. In alignment with the “One Health approach” policy, which underscores the interconnectedness of human, animal, and environmental health, a focused inquiry into the biodiversity of nature, particularly within the realm of plant-derived compounds, emerges as a promising avenue. This study endeavors to contribute to the pursuit of novel antimicrobial agents and antioxidants, with a specific focus on combating prevalent afflictions such as dental caries. It involves leveraging the wisdom of traditional medicine and combining it with modern scientific methodologies to unlock the full potential of natural substances. Within the context of the current investigation, diverse extracts derived from two distinct plants, namely *R. damascene* and *H. perforatum*, were subjected to scrutiny to elucidate their biological activities and antibacterial efficacy against members of both oral microbiota and foodborne pathogens. This exploration is rooted in the recognition of the profound impact that dietary habits exert on the prevalence of dental caries and overall oral health.

### 4.1. The Chemistry and Antioxidant Activities of Plant Extracts

#### 4.1.1. General Aspects

In this study, phytochemical screening involves qualitatively analyzing plant extracts to identify various classes of secondary metabolites. Both plants, *R. damascene* and *H. perforatum*, exhibit richness in flavonoids, alkaloids, terpenoids, tannins, and glycosides. Notably, *H. perforatum* extracts surpass *R. damascene* in anthraquinones. Enzymatic extracts of both species reveal the presence of all phytochemical groups, indicating that enzymatic extraction, facilitated by biological catalysts (enzymes), enhances the recovery of certain compounds. This method proves particularly valuable for accessing bioactive compounds that may be challenging to obtain through other means, as also discussed elsewhere [42,72]. Moreover, enzymatic methods can be selective, yielding extracts enriched with specific phytochemical classes. The choice of extraction method depends on the compounds of interest, the plant’s characteristics, and geographical factors [73,74]. The specific phytochemical profile is influenced by the plant species’ “fingerprint”, including geographical variation, environmental factors, temperature, altitude, humidity, and light. Additionally, it is essential to note that the phytochemicals detected depend on the extraction method and analytical techniques, as also described elsewhere [73,74,75].

Terpenoids, derived from isoprene units, constitute a diverse group of organic compounds produced by plants, fungi, and some animals. Their biological activities encompass antimicrobial, anti-inflammatory, antioxidant, analgesic, wound healing, and anticancer effects. The pharmacological properties of a terpenoid are influenced by its chemical structure, concentration in the organism, and interactions with biological targets. Synergistic effects within plant extracts contribute to their overall pharmacological profile [76,77]. Tannins, exhibiting similar biological actions, play a role in maintaining oral hygiene. They possess astringent properties, inhibit enzymes (including those from oral bacteria), and reduce plaque and tartar formation. Moreover, they contribute to cavity prevention by inhibiting cariogenic bacteria and preventing tooth enamel demineralization [78,79].

Certain glycosides, such as quercetin and its glycosides (e.g., quercetin-3-O-rutinoside in *R. damascene*), are studied for their antimicrobial, anti-inflammatory, and antioxidant properties. Quercetin glycosides, prevalent in many flowers and fruits, play a crucial role in preventing the release of pro-inflammatory mediators [80,81]. Chroho et al., conducted research on the composition of extracts from flowers of Rosa damascene from Morocco and found that Quercetin and kaempferol, with their derivatives and glycosides (as kaempferol-3-O-rutinoside), were the major detected flavonoids, with kaempferol derivatives predominating as an isolated component [82].

In our study, all *H. perforatum* extracts were richer in anthraquinones, a finding in accordance with previous studies, which, logically, are in the form of the anthraquinones emodin and/or emodin anthraquinone, and are involved in the hypericin biosynthetic pathway (bioactive compound found in some species of the genus Hypericum, commonly known as St. John’s Wort).

#### 4.1.2. Total Phenolic and Flavonoid Content

Phenolic compounds, widely distributed in plants, exhibit diverse biological activities owing to their redox properties, allowing them to function as antioxidants. Flavonoids, a subset of phenolic compounds, demonstrate antibacterial, anti-inflammatory, wound healing, antiulcer, hepatoprotective, anticancer, and neuroprotective activities, emphasizing their role in combating oxidative stress and supporting antioxidant defense mechanisms. Ongoing research highlights their therapeutic potential across various health conditions [73,83]. In *R. damascene*, the aqueous extract showed the highest total phenolic content (TPC) at 104.92 ± 6.05 mg GAE/g, followed by the aqueous/methanolic fractions (89.1 ± 2.2), with the 60% ethanolic sample exhibiting the lowest TPC (75.47 ± 1.79 mg GAE/g). This aligns with the principle that polar solvents are more effective for extracting polar compounds, reflected in the increasing phenolic content from ethanolic to water extracts across plant species. Recent research indicates that the phenolic compound composition in *R. damascene* extracts varies with the sample’s origin, showing an increasing trend from aqueous to hydro-methanolic extracts. These findings underscore the wide variability in polyphenols based on factors such as plant species, plant part, environmental conditions, and extraction methods [82,84].

Also, a significant difference was observed in the resulting phenolic compounds for the four different solvent-based extracts in total flavonoid content within the same plant, with the following descending order of recorded values: enzymatic extract (42.78 ± 0.41 mg CE/g), aqueous extracts (41.97 ± 0.34 mg GAE/g), 40% ethanolic extract (28.73 ± 0.45), and 60% ethanolic extract (20.68 ± 0.5 mg CE/g). The interpretation of these results in comparison with other studies cannot be conducted, because until today no enzymatic method of extraction from *R. damascene* has been used. Researchers studying the antioxidant profiles of commonly consumed edible flowers in China recorded values similar to those of our study, with the only difference being that an acetone/water/acetic acid- based solvent was used (China rose, 24.13 mg catechin equivalents, CAE/g; rose, 23.56 mg CAE/g) [85]. In previous works using a 70% ethanolic based solvent for the extraction, the amount of total flavonoids has been reported at 28.1–98 mg and 28.59 mg, expressed in the units of catechin equivalents/DW and Qu/g DW, respectively [86,87].

In the case of *H. perforatum* extracts, a different pattern of total phenolic values is captured, with the highest value recorded in the 60% ethanolic extract (92.39 ± 2.06 mg GAE/g) and the lowest in the enzymatic extract (49.2 ± 0.83 mg GAE/g), while in the case of total flavonoids, the superiority of the enzymatic extract remains (33.63 ± 1.3 mg GAE/g). Alahmad et al. showed with their results that phenolic compounds were present in lower concentrations in ethanolic extracts (64.4 mg GAE/g), when compared to the methanolic extracts (93.2 mg GAE/g), and in even lower concentrations when compared to the aqueous extracts (170.6 ± 1.7 mg GAE/g), stating as a possible justification the fact that methanolic and ethanolic extracts do not dissolve fully in water [88]. The total flavonoid content of ethanol and ethanol–water extracts prepared either from air-dried samples of aerial parts of *H. perforatum* or from lyophilized material, reported from Makarova et al., is much higher than the total flavonoid content recorded in our study for the extracts prepared from dried and lyophilized flowers [89]. However, it is very important to state the fact that the enzymatic extracts display a satisfactory value of total flavonoids.

#### 4.1.3. The Evaluation of Antioxidant Capacity: DPPH Assay and Ferric Reducing Assay Power (FRAP)

The DPPH (2,2-diphenyl-1-picrylhydrazyl) radical scavenging assay, a commonly used method to evaluate the antioxidant activity of various compounds, including medicinal plant extracts, was applied in the present study. This assay is based on the ability of antioxidants to donate electrons or hydrogen atoms to reduce the stable DPPH radical to a non-radical form, into the reduced form DPPH·-H. The extent of DPPH radical reduction is measured spectrophotometrically, and a decrease in absorbance indicates a higher free radical scavenging activity. In our study, the aqueous extracts of *R. damascene* showed the highest values of scavenging activity, following by ethanolic and enzymatic extracts (Table 3). This is perfectly consistent with the fact that the aqueous extracts had recorded high values of total phenolics content, strengthening the hypothesis expressed by researchers in previous studies, that the difference in the free radical scavenging activity of extracts is based on their chemical composition and content of total phenols and flavonoids [88,90]. Despite variations in extraction methods, all extract types exhibited noteworthy antioxidant capacity. Our findings, however, diverge from previous studies, particularly those focused on *R. damascene* extracts, wherein ethanolic or methanolic extracts were reported to have higher antioxidant capacities compared to aqueous extracts [91,92,93,94]. This incongruity underscores the influence of chemical composition on the antioxidant capacity of extracts. In the instance of *H. perforatum*, the 40% and 60% ethanolic extracts exhibited the highest antioxidant capacity in DPPH assays. An examination of Table 2 reveals a certain pattern, with ethanolic extracts consistently demonstrating higher total phenolic content. This recurrence parallels the findings in the case of *R. damascene*. Notably, prior research has contradicted our results, indicating lower antioxidant capacity values for ethanolic or methanolic extracts of *H. perforatum* compared to aqueous extracts [88,95]. Nonetheless, a consensus exists regarding the strong correlation between phenolic content, antioxidant activity, and the extraction medium. The choice of solvent, whether it be methanol (MeOH), ethanol (EtOH), water, or other solvents, can significantly impact the composition of extracted compounds, particularly phenolic compounds. Table 4 shows the dose–response values for the reducing powers of all extracts (25–250 μg/mL). The data indicate that increased extract concentrations resulted in higher reducing power in both plants. Regarding *R. damascene*, the descending order in terms the values of reducing power were as follows: ethanolic extracts (40% and 60%), enzymatic extracts, and finally, aqueous extracts. In contrast, in the case of *H. perforatum,* the top of the descending ranking is held by the enzymatic extract, followed by the aqueous extracts, and ending with the ethanolic extracts (40% and 60%).

### 4.2. The Antimicrobial and Antioxidant Capacity of the Studied Extracts

The oral microbiota, a diverse microbial community within the oral cavity, plays a pivotal role in maintaining oral health, with imbalances contributing to various oral diseases. Understanding this micro-ecosystem extends beyond oral health, impacting broader systemic health. Key characteristics include a polymicrobial ecosystem, a dynamic balance influenced by factors like diet and oral hygiene, a delicate equilibrium between pathogens and commensals, biofilm formation (exemplified by dental plaque), and implications for systemic health, with links to conditions such as cardiovascular disease and diabetes [94,95,96]. Unlike gastrointestinal microbiota, oral microbial communities exhibit minimal changes due to diet and environment, except for an increased diversity in caries and periodontal diseases [96,97,98]. In exploring the bidirectional axis of homeostasis–dysbiosis in the oral cavity, crucial elements involve manipulating microbial communities and interactions within the host, emphasizing targeted preventive measures for overall oral health. In considering inflammation’s role in dysbiosis, periodontitis-associated inflammophilic bacteria, like *P. gingivalis*, manipulate the host immune response, creating a dual approach for managing oral infections: addressing inflammation while enhancing antimicrobial capacity [96,99,100]. Previous discussions on the anti-inflammatory properties of plants set the stage for the second part of this approach.

The antimicrobial efficacy of plants pertains to their capacity to impede the proliferation of, or eradicate, microorganisms, encompassing bacteria, fungi, viruses, and other pathogens. Plants have evolved an array of chemical compounds as the constituents of their defense mechanisms against microbial infections. These bioactive compounds, including alkaloids, flavonoids, tannins, essential oils, and other secondary metabolites, manifest antimicrobial properties, rendering them prospective reservoirs for the advancement of novel antimicrobial agents or drugs [75,101].

The disk diffusion test didn’t show a clear antibacterial pattern, but it did indicate that these extracts work similarly to antibiotics. The antibacterial effect was dependent on the bacterial species, on the type of the extract (in most cases E40 and E60 were more effective than the other extracts for *R. damascene* while the enzymatic extracts seemed to have more powerful antibacterial properties than the rest of the extracts for *H. perforatum*), and on the extract content of the disk (Table 5 and Table 7). For instance, *F. nucleatum* appeared to be the most sensitive microorganism tested against *R. damascene* extracts and this observation is important due to the clinical significance of this bacterium, since it is known to migrate from the oral cavity and cause serious infections in the heart (e.g., endocarditis). For *H. perforatum* extracts, *P. intermedia* was the most sensitive of all. In general, all microorganisms tested showed susceptibility to the extracts and this effect was very strongly dose-responsive (all Spearman coefficients had values greater than 0.842) and proportional to the concentration of the extracts in the disks (Table 6 and Table 8).

Figure 3 demonstrates the comparative antibacterial potency of the four extracts of each plant against each other and against 12 clinically used antibacterial substances in the means of the diameters of the inhibitory zones. Considering the variation, one could argue that the extracts do have a similar antibiotic activity to the pharmaceutical substances. Given the fact that the extracts contain unknown substances in unknown concentrations, their effect could be synergistic or even antagonistic. In the latter case, the possibility of the presence of a powerful substance whose action is inhibited by another substance cannot be excluded, and this is a serious limitation of this study. Table 1 shows the phytochemical content of the extracts and Table 2 shows the flavonoid and phenolic substance content of the same extracts. These tables reveal the richness of these extracts in pharmacologically active substances. Furthermore, these substances are present in different concentrations depending on the solvent, hence the different potency of each mixture. For the same reasons, MIC and MBC values differ among the extracts. As with the inhibitory zones, here too these values depend on the strain and on the type of the extract. For example, the enzymatic extracts of *H. perforatum* show lower MIC and MBC values for all of the strains tested while *P. gingivalis* required the lowest MIC and MBC values for the aqueous extract (Table 10). *F. nucleatum* and *P. intermedia* required the lowest MIC and MBC values for the aqueous extract of *R. damascene* (Table 10).

Performing a bibliometric overview on the study of *R. damascene* extracts against pathogenic bacteria, we notice that *S. aureus* is the most studied bacterium, against which it is confirmed that various types of extracts, including aqueous, ethanolic, and methanolic extracts, show strong antimicrobial activity [102,103]. This is the reason that we also incorporated in *S. aureus* our study. Other studies also showed that *R. damascena* has a bacteriostatic or bactericidal activity on cariogenic bacteria *(S. mutans, Streptococcuss anguinis* and *Streptococcus sobrinus*) and on bacteria that are the causative agent of periodontal diseases, such as *P. gingivalis*, *Actinobacillus* sp., *Prevotella* sp., and *Fusobacterium* sp. [104,105,106,107,108].

According to the results of previous studies, *H. perforatum* oil or solvent-based extracts have antibacterial and anti-biofilm properties against the common bacteria associated with periodontitis, such as *P. gingivalis*, *Escherichia coli*, *S. mutans*, *S. sobrinus*, and *S. aureus* Furthermore, Bagheri et al. showed that *H. perforatum* oil has the same bactericidal ability as that shown by the antibiotic control groups [88,109,110,111,112,113].

Antibiofilm effectiveness is imperative in antibacterial combat since biofilms, through their physical and chemical properties, protect the participating microorganisms. As it can be seen in Figure 2, the ethanolic, the enzymatic, and the aqueous/ethanolic extracts of *H. perforatum* showed an absolutely excellent antibiofilm score, higher than the ones of the conventional clinical antibiotics. Similar results, though less potent but still outperforming the conventional antibiotics, were seen in most cases with the (aqueous and enzymatic) extracts of *R. damascene* (Table 12 and Figure 3).

Time-kill curves (Figure 5) provide the necessary graphic pattern to understand the potency of any bactericidal substance. The relation is strain-specific, since three strains appear to be more resistant to the aquatic extract of *R. damascene*. The MRSA/VRSA, the MRSA from raw milk, and the MRSA from raw broiler carcass remained almost unharmed for the first 12 h, and after that period they started dying in proportion to the MBC concentration. Practically, only eight-times the MBC concentration managed to eliminate them within 24 h. Perhaps these strains had acquired resistance mechanisms due to an environmental origin. The proportionality however, between MBC concentration and bacterial death can be observed in all tested strains in the current research, revealing a dose-responsive effect. The oral pathogens, as well as the reference strains, were more susceptible. The former were eliminated within 8–12 h in concentrations higher than twice the MBC, while the reference strain population was significantly reduced to 1 log/cfu in the first 12 h and eliminated over the next 12 h of contact (Figure 4).

An interesting observation can be derived from Table 14. The aquatic extract is less effective than the enzymatic one (in hours necessary to eliminate the bacteria) in the MRSA/VRSA, the MRSA from raw milk, and the MRSA from raw broiler carcass, while the opposite observation is true for the other bacteria. This result pinpoints the fact that the pattern of susceptibility is dependent on the strain. Moreover, it seems that extracts may easily penetrate Gram (+) bacteria due to the relatively thick, but simpler cell wall structure. Gram (−) bacteria, with their more complex cell wall arrangement, may present a greater challenge for substances to penetrate [96,114]. However, it is essential to note that the effectiveness of extracts depends on various factors, including the specific constituents of the extracts and the mechanisms of action involved [99,115].

*R. damascena,* with its diverse bioactive compounds like neral, geranial, phenyl ethyl, and phenolic compounds, exhibits inhibitory effects on microbial growth, targeting bacterial DNA gyrase and co-enzymes crucial for bacterial survival. Notably, phenolic compounds, including kaempferol and quercetin, showcase potential antimicrobial properties, making *R. damascena* a subject of interest for natural antimicrobial and anti-inflammatory agents [82,84,116]. Decoctions, essential oil, and absolute, methanol, and ethanol extracts of rose petals have exhibited antioxidant activity in different systems [117,118], as well as antimicrobial activity against *S. aureus*, *S. typhimurium*, *B. cereus*, *C. albicans*, *P. aeruginosa*, *P. fluorescens*, etc. [119,120,121,122,123,124,125,126]. The efficacy of herbal mouthwash containing aqueous rose extract in the treatment of recurrent aphthous stomatitis has also been reported [127]. The antioxidant activities, including DPPH, ABTS, and FRAP, of different rose extracts have been already reported [128,129,130,131], but it is difficult to compare the results due to differences in the assay procedures, or in the solvents used for extraction. The synergistic action of these compounds leads to beneficial effects, and this is the first step towards understanding the mechanism of action. Polyphenols have also been proposed as antioxidants and scavengers of peroxyl and superoxide radicals and have a role in the control of excessive reactive oxygen species (ROS). Polyphenols’ antioxidant properties have also been shown to protect against chronic inflammation [132]. Polyphenols have been shown to regulate transcription factors involved in lipid and glucose homeostatic metabolism (e.g., AMPK, PPARs, and SREBP-1c) [132]. Polyphenols’ modes of action have been known for having pleiotropic consequences which might involve signal transduction pathways [133].

*H. perforatum*, extensively studied for its antibacterial properties, boasts compounds like hyperforin, hypericin, and pseudohypericin, with antibacterial mechanisms involving DNA inhibition, cell membrane disruption, enzyme modulation, and antioxidant effects [134,135,136,137]. Despite promising antibacterial results, the documented development of resistance in microorganisms to these extracts, linked to antibiotic exposure over time, emphasizes the need for caution [75,82,138]. The oral cavity, a complex ecosystem, relies on a delicate microbial balance for health. Our study provides valuable insights into *R. damascena* and *H. perforatum* extracts’ phytochemical composition, antioxidants, and anti-inflammatory actions, paving the way for potential antibiotic biomimics to address oral health challenges, including dental caries [108,139,140]. Antibacterial and anti-inflammatory properties, as well as the enhancement of fibroblast movement and collagen formation, are all factors in how *H. perforatum* treats wounds [141,142]. *H. perforatum* has a lot of bioactive compounds with anti-inflammatory properties, and more recently, it has primarily been used to treat anxiety and depression in place of traditional antidepressants, with which it shares the inhibition of the uptake of monoamine neurotransmitters [143,144,145,146]. The whole extract of *H. perforatum* frequently contains other active substances such hyperoside, rutin, quercetin, and various catechins, though their concentrations might vary greatly depending on seasonal variations and the plant’s place of origin [88,147,148].

## 5. Limitations of the Study

Despite the promising findings, several limitations must be considered. First, the study acknowledges the potential variability in the antimicrobial and antioxidant efficacy of the extracts. This variability arises from factors such as the extraction method, solvent used, and geographical variations, introducing a level of unpredictability in the observed outcomes [73,74,75]. Second, the presence of unknown substances in the extracts poses a challenge in understanding potential synergistic or antagonistic effects among the compounds. Thus, we emphasize the need for caution in interpreting the overall impact of the extracts due to these uncertainties [102]. Additionally, the strain-specific responses observed in the study’s time-kill curves highlight the complexity of microbial interactions, indicating that certain strains may exhibit higher resistance or require longer exposure times [114]. The documented development of resistance in microorganisms to these extracts, possibly linked to prolonged antibiotic exposure, underscores the importance of ongoing monitoring and careful consideration of the long-term effectiveness of these natural compounds [121]. Lastly, further research is needed on the chemical analysis of the plant compounds, as well as of the extracts to fully detect the substances involved thus postulate conclusions on their mechanisms of action.

Despite these limitations, the study’s insights hold significant implications for future clinical applications. Firstly, the research sets the stage for the potential development of novel antimicrobial agents or drugs derived from natural compounds found in *R. damascena* and *H. perforatum*. These alternatives could offer innovative solutions for the treatment of oral diseases and contribute to the global effort against antibiotic resistance [75,101]. Moreover, the understanding of the phytochemical composition and antibacterial properties of the studied extracts provides a foundation for the development of oral health formulations. Integration into oral care products could address issues such as dental caries and promote overall oral health [108,122,123]. Furthermore, the diverse biological activities of terpenoids, tannins, glycosides, and other compounds present in the extracts open avenues for exploring synergistic therapeutic approaches. Combinations of natural compounds could be investigated for enhanced antimicrobial, anti-inflammatory, and antioxidant effects in the context of oral health applications [76,77]. Additionally, the demonstrated antibiofilm effectiveness of the extracts, especially against oral pathogens, suggests potential applications in biofilm management. Future research could then delve into the development of strategies targeting biofilm-associated oral diseases [96,99,100]. To validate the clinical efficacy and safety of these natural extracts, future research should identify specific phenolic and flavonoid compounds, optimize extraction parameters, explore direct interactions with oral microbiota, and conduct clinical trials for validation. So future research approaches could involve well-designed clinical trials, providing valuable data on the real-world effectiveness of these compounds and assessing any potential side effects or adverse reactions. Ultimately, the study’s emphasis on preventive aspects aligns with the evolving paradigm of holistic oral healthcare, encouraging the integration of plant-based therapies into preventive oral health practices for sustainable and environmentally conscious approaches [35,36].

## 6. Conclusions

This study offers promising insights into the phenolic and flavonoid content, as well as the antioxidant capacity, of extracts from *Rosa damascena* and *Hypericum perforatum* against oral microbiota. However, acknowledging the limitations, such as variability in efficacy due to extraction methods and unidentified substances in the extracts, is crucial for cautious interpretation. The strain-specific responses and the potential development of resistance in microorganisms underscores the need for ongoing monitoring. Despite these limitations, the study has implications for future clinical applications, paving the way for the development of novel antimicrobial agents derived from natural compounds. This could contribute to addressing oral diseases and combating antibiotic resistance. Understanding the extracts’ phytochemical composition provides a foundation for potential oral health formulations, addressing diseases like dental caries. The biological activities of compounds in the extracts offer opportunities for exploring synergistic therapeutic approaches, enhancing antimicrobial, anti-inflammatory, and antioxidant effects. Additionally, the demonstrated antibiofilm effectiveness suggests potential applications in biofilms. The following are particularly important in oral pathogen management:-The antibacterial effect of the studied plants extracts against oral pathogens was absolute in the kill-time kinetics, implying an increased antibacterial potential of these extracts.-Certain differences in the time-kill kinetics curve can be attributed to species-specific factors.-A similar elimination of the bacterial cells was also observed in the case of the foodborne bacteria, in time-kill kinetics.

Future research should focus on validating clinical efficacy, identifying specific compounds, optimizing extraction parameters, exploring direct interactions with oral microbiota, and conducting well-designed clinical trials. This holistic approach aligns with the evolving paradigm of integrating plant-based therapies into preventive oral health practices for sustainable and environmentally conscious approaches.

## Figures and Tables

**Figure 1 microorganisms-12-00060-f001:**
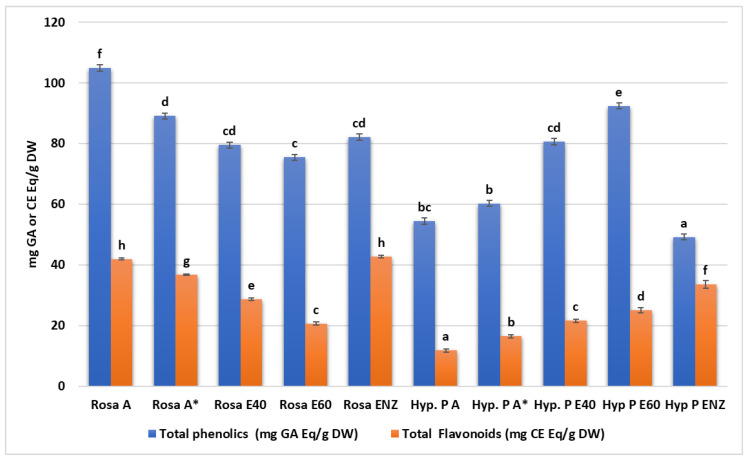
Total phenolics (mg GA Eq/d DW) and total flavonoids (mg CE Eq/g DW) among the various extracts of *R. damascene* and *H. perforatum*. Similar letters over bars indicate no statistically significant differences (Kruskal–Wallis, *p* > 0.05 with Tukey’s HSD).

**Figure 2 microorganisms-12-00060-f002:**
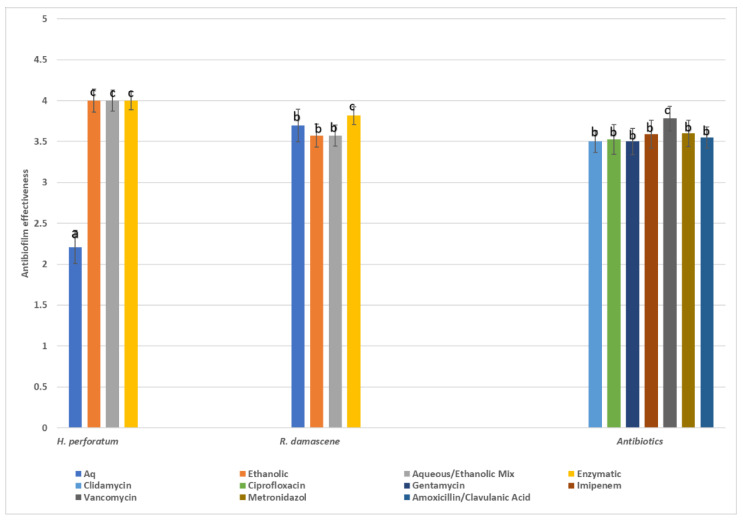
Total median values (score 1: poor, 2: good, 3: very good, 4: excellent) of the antibiofilm effectiveness of the various types of extracts from the two herbals. Values close to 1 indicate poor biofilm inhibition (0–50%), while values close to 4 indicate excellent biofilm inhibition (>95%). Similar letters above bars indicate no significant difference in antibiofilm effectiveness between similar types of extracts among the various herbal species and antibiotics (Kruskal–Wallis, *p* < 0.05).

**Figure 3 microorganisms-12-00060-f003:**
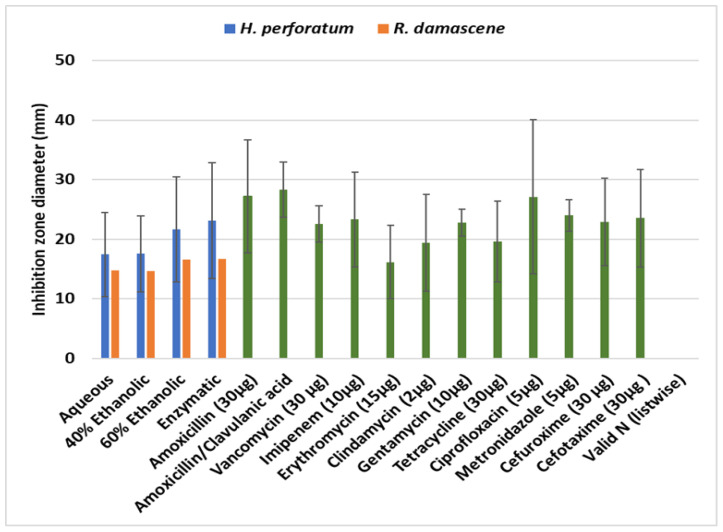
Mean inhibition zones (±standard deviation) of the various herbal extracts and antibiotics against oral and other pathogens.

**Figure 4 microorganisms-12-00060-f004:**
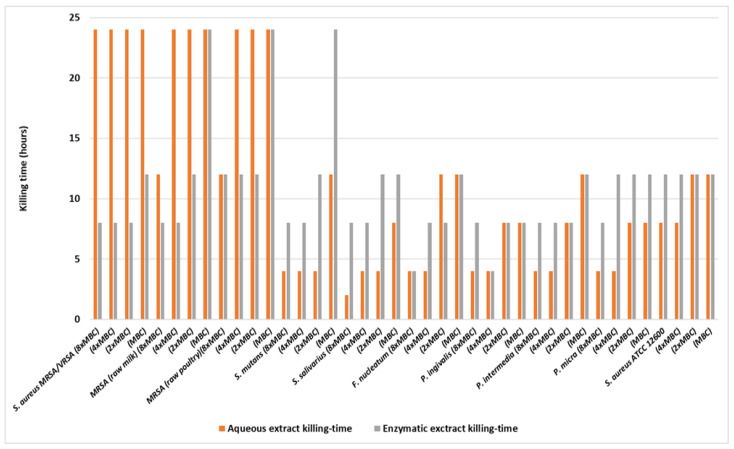
Killing time (in hours) of the various concentrations (1×, 2×, 4×, 8× Minimum Bactericidal Concentration) of the aqueous and enzymatic extracts of *R. damascene* against the various pathogens.

**Figure 5 microorganisms-12-00060-f005:**
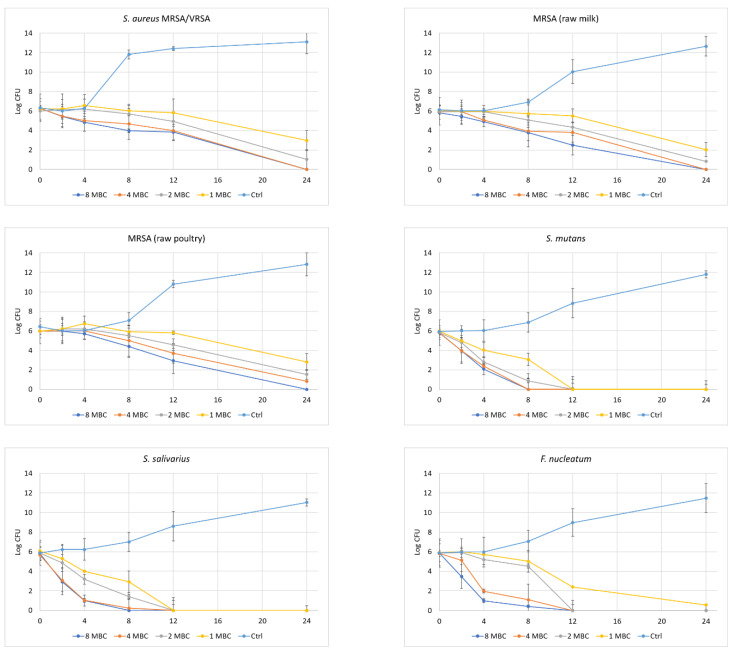
Time-kill curves of various concentrations of aqueous (Aq) extracts from *R. damascene* against 9 pathogens and 1 reference strain. X-axis presents the contact time in hours.

**Table 1 microorganisms-12-00060-t001:** Phytochemical constituents are present in individual plant extracts.

Phytochemical Compounds	Plants Extracts							
	Roses (*Rosa damascene*)				*Hypericum perforatum*			
	A	Ε40	Ε60	ΕΝΖ	A	Ε40	Ε60	ΕΝΖ
Alkaloids	+	+	+	+	+	+	+	+
Anthraquinones	−	−	−	+	+	+	+	+
Terpenoids (Salkowski’s test)	+	+	+	+		+	+	+
Steroids	+	+	+	+	−	−	−	−
Saponins	+	−	−	+	−	−	−	+
Flavonoids (alkaline reagent test)	+	+	+	+	+	+	+	+
Tannins (ferric chloride test)	+	+	+	+	+	+	+	+
Glycosides (Keller–Kiliani test)	+	+	+	+	+	+	+	+

A—aqueous extract, E40—ethanolic extract (40% *v*/*v* aqueous ethanol), E60—ethanolic extract (60% *v*/*v* aqueous ethanol), ENZ—enzymatic extract; +: positive detection, −: negative detection.

**Table 2 microorganisms-12-00060-t002:** Total phenolics and flavonoid content of plant extracts.

Plant Extracts	Total Phenolics (mg Gallic Acid Equivalent/g of Dried Sample)	Total Flavonoids (mg Catechin Equivalent per/g of Dried Sample)
*R. damascene*		
Rosa A	104.92 ± 6.05 ^c^	41.97 ± 0.34 ^d^
Rosa A*	89.1 ± 2.2 ^b^	36.81 ± 0.26 ^c^
Rosa E40	79.47 ± 2.38 ^a^	28.73 ± 0.45 ^b^
Rosa E60	75.47 ± 1.79 ^a^	20.68 ± 0.5 ^a^
Rosa ENZ	82.12 ± 2.95 ^ab^	42.78 ± 0.41 ^d^
*H. perforatum*		
Hyp. P A	54.45 ± 3.03 ^ab^	11.78 ± 0.62 ^a^
Hyp. P A*	60.26 ± 0.94 ^b^	16.47 ± 0.58 ^b^
Hyp. P E40	80.61 ± 6.85 ^c^	21.56 ± 0.62 ^c^
Hyp P E60	92.39 ± 2.06 ^d^	25.06 ± 0.87 ^d^
Hyp P ENZ	49.2 ± 0.83 ^a^	33.63 ± 1.3 ^e^

Values are the mean of three replicates (TPC). Different letters in the columns denote statistical differences between the total phenolics or total flavonoids among the various extracts in each plant (Kruskal–Wallis, *p* < 0.05 with Tukey’s HSD).

**Table 3 microorganisms-12-00060-t003:** Percentage of Neutralization of the DPPH Radical by Plant Extracts in the DPPH Assay.

Plant Extracts	Concentration (μg/mL)					
	500	300	100	50	10	1
*R. damascene*						
Rosa A	86.05 ± 0.14 ^c^	83.26 ± 0.26 ^b^	72.32 ± 0.4 ^d^	45.09 ± 0.9 ^d^	12.85 ± 0.27 ^d^	4.3 ± 0.25 ^d^
Rosa A*	85.11 ± 0.12 ^c^	85.41 ± 0.96 ^b^	62.13 ± 0.82 ^c^	43.09 ± 0.54 c	11.26 ± 0.55 ^c^	3.84 ± 0.25 ^d^
Rosa E40	65.85 ± 0.53 ^b^	61.18 ± 0.24 ^ab^	50.68 ± 0.15 ^b^	41.93 ± 0.57 ^bc^	14.05 ± 0.19 ^e^	2.96 ± 0.1 ^c^
Rosa E60	64.45 ± 0.37 ^a^	42.49 ± 31.3 ^a^	39.32 ± 0.3 ^a^	28.07 ± 0.17 ^a^	7.67 ± 0.56 ^a^	1.99 ± 0.54 ^b^
Rosa ENZ	65.03 ± 0.88 ^ab^	59.4 ± 1.02 ^ab^	51.33 ± 0.57 ^b^	41.31 ± 0.57 ^b^	9.22 ± 0.32 ^b^	0.87 ± 0.11 ^a^
*H. perforatum*						
Hyp. P A	61 ± 1.4 ^b^	58.6 ± 0.86 ^b^	39.51 ± 0.68 ^c^	31.53 ± 0.6 ^b^	10.65 ± 0.22 ^c^	3.5 ± 0.42 ^b^
Hyp. P A*	67.87 ± 0.45 ^c^	53.11 ± 0.77 ^a^	37.32 ± 0.4 ^b^	24.02 ± 0.49 ^a^	3.38 ± 0.39 ^a^	0.67 ± 0.24 ^a^
Hyp. P E40	87.95 ± 0.36 ^d^	82.81 ± 0.77 ^c^	77.16 ± 0.25 ^d^	43.66 ± 0.76 ^c^	11.05 ± 0.19 ^c^	4.22 ± 0.29 ^b^
Hyp P E60	89.16 ± 0.37 ^d^	87.44 ± 0.48 ^d^	78.99 ± 0.12 ^e^	54.43 ± 0.57 ^d^	16.66 ± 0.36 ^d^	11.32 ± 0.48 ^c^
Hyp P ENZ	55.29 ± 0.52 ^a^	52.38 ± 0.5 ^a^	29.02 ± 0.48 ^a^	25.12 ± 0.46 ^a^	6.15 ± 0.09 ^b^	1.42 ± 0.16 ^a^

Values are the mean of three replicates. Different letters in concentration columns denote statistical differences among the various extracts in each plant (Kruskal–Wallis, *p* < 0.05 with Tukey’s HSD).

**Table 4 microorganisms-12-00060-t004:** Reducing power of different concentrations of extracts from *R. damascene* and *H. perforatum,* as evaluated using the FRAP assay.

Plant Extracts	Concentration (μg/mL)					
	25	50	100	150	200	250
*R. damascene*						
Rosa A	0.18 ± 0 ^a^	0.26 ± 0.01 ^a^	0.32 ± 0.01 ^a^	0.75 ± 0 ^a^	0.91 ± 0.01 ^a^	1.02 ± 0 ^a^
Rosa A*	0.2 ± 0 ^b^	0.34 ± 0 ^b^	0.68 ± 0.01 ^c^	0.88 ± 0.01 ^b^	1 ± 0.01 ^b^	1.22 ± 0.01 ^b^
Rosa E40	0.2 ± 0 ^b^	0.48 ± 0 ^d^	0.68 ± 0.01 ^c^	1.18 ± 0 ^c^	1.31 ± 0.01 ^d^	1.46 ± 0 ^c^
Rosa E60	0.21 ± 0 ^c^	0.49 ± 0.01 ^d^	0.71 ± 0 ^d^	1.22 ± 0 ^d^	1.35 ± 0 ^e^	1.54 ± 0.01 ^e^
Rosa ENZ	0.18 ± 0 ^a^	0.35 ± 0.01 ^c^	0.65 ± 0 ^b^	1.18 ± 0.01 ^c^	1.22 ± 0 ^c^	1.47 ± 0.01 ^d^
*H. perforatum*						
Hyp. P A	0.5 ± 0.57 ^d^	0.28 ± 0.01 ^b^	0.35 ± 0 ^b^	0.74 ± 0.01 ^b^	1.02 ± 0.01 ^b^	1.22 ± 0 ^b^
Hyp. P A*	0.23 ± 0.01 ^c^	0.44 ± 0.01 ^c^	0.79 ± 0.01 ^c^	0.92 ± 0.01 ^c^	1.22 ± 0.01 ^c^	1.44 ± 0.01 ^c^
Hyp. P E40	0.21 ± 0.01 ^b^	0.22 ± 0 ^a^	0.32 ± 0.01 ^a^	0.44 ± 0 ^a^	0.65 ± 0 ^a^	0.89 ± 0.01 ^a^
H-yp P E60	0.19 ± 0.01 ^a^	0.22 ± 0.01 ^a^	0.32 ± 0.01 ^a^	0.45 ± 0 ^a^	0.66 ± 0.01 ^a^	0.88 ± 0.01 ^a^
Hyp P ENZ	0.23 ± 0 ^a,c^	0.52 ± 0.01 ^d^	0.78 ± 0.01 ^c^	1.33 ± 0.01 ^d^	1.45 ± 0.01 ^d^	1.61 ± 0.01 ^d^
Reference						
Gallic acid (GA)	0.18 ± 0.01	0.43 ± 0.01	1.82 ± 0.01	1.85 ± 0.01	1.88 ± 0	1.88 ± 0.01
Ascorbic acid (AA)	0.19 ± 0	0.42 ± 0.01	0.69 ± 0.01	1.13 ± 0.01	1.66 ± 0.01	1.89 ± 0.01

Values are the mean of three replicates. Different letters in concentration columns denote statistical differences among the various extracts in each plant (Kruskal–Wallis, *p* < 0.05 with Tukey’s HSD).

**Table 5 microorganisms-12-00060-t005:** Antibacterial activities (disk diffusion) of various concentrations (10, 20, 50 and 100%) of aqueous (A), ethanolic (E40 & E60), and enzymatic (Enz) extracts from Hypericum perforatum against oral pathogens and a reference strain.

Pathogen	Disk Content (%)	A (mm)	E40 (mm)	E60 (mm)	Enz (mm)
*S. aureus* MRSA/VRSA	10	10.7 ± 0.7 ^a^	12.2 ± 0.3 ^a^	14.3 ± 0.9 ^b^	14 ± 0.8 ^b^
	20	14.6 ± 0.4 ^a^	14.9 ± 0.2 ^a^	14.2 ± 0.3 ^a^	21.9 ± 0.6 ^b^
	50	17.5 ± 0.6 ^a^	21.5 ± 0.6 ^a^	18.4 ± 0.6 ^b^	31.4 ± 1.1 ^c^
	100	23.1 ± 2 ^a^	25 ± 5.2 ^ab^	31.7 ± 0.6 ^b^	48.2 ± 1 ^c^
MRSA (raw milk)	10	9.4 ± 0.4 ^a^	8.3 ± 0.4 ^a^	12.4 ± 0.4 ^b^	12.2 ± 0.7 ^b^
	20	15 ± 1.6 ^a^	12.2 ± 0.4 ^ab^	14.1 ± 0.6 ^b^	15.7 ± 0.8 ^b^
	50	17.9 ± 0.1 ^a^	15 ± 0.3 ^b^	18.6 ± 0.5 ^b^	20.5 ± 0.6 ^c^
	100	21.7 ± 0.8 ^a^	19.5 ± 0.6 ^b^	22.8 ± 1.2 ^b^	29.2 ± 0.2 ^c^
MRSA (raw poultry)	10	11.2 ± 1 ^a^	11.7 ± 0.4 ^ab^	12.9 ± 0.3 ^b^	10.5 ± 0.4 ^a^
	20	17.3 ± 0.5 ^b^	15.8 ± 0.7 ^a^	18 ± 0.2 ^b^	20.5 ± 0.6 ^c^
	50	28.2 ± 0.9 ^c^	24.9 ± 1.3 ^ab^	24.8 ± 0.7 ^a^	27.1 ± 0.2 ^bc^
	100	35.2 ± 0 ^b^	30.5 ± 0.5 ^a^	30.1 ± 0.7 ^a^	40.7 ± 0.5 ^c^
*S. mutans*	10	9.2 ± 0.6 ^a^	11 ± 0.7 ^b^	12.1 ± 0.8 ^bc^	13.7 ± 0.4 ^c^
	20	13.4 ± 0.5 ^a^	12 ± 0.4 ^a^	15.1 ± 0.3 ^b^	20.1 ± 0.9 ^c^
	50	18.3 ± 0.6 ^a^	18.9 ± 6 ^a^	20.4 ± 0.4 ^a^	30.4 ± 0.5 ^b^
	100	26.6 ± 1.5 ^a^	22.4 ± 5.6 ^a^	28.6 ± 0.6 ^a^	37.9 ± 0.4 ^b^
*S. salivarius*	10	10.7 ± 0.6 ^a^	11.9 ± 0.6 ^ab^	13.3 ± 0.4 ^b^	12.6 ± 0.5 ^b^
	20	14.6 ± 0.5 ^a^	14.4 ± 0.6 ^a^	19.1 ± 0.1 ^b^	18.2 ± 0.3 ^b^
	50	19 ± 0.1 ^a^	20.7 ± 0.5 ^a^	28 ± 0.2 ^b^	27.4 ± 0.8 ^b^
	100	24.3 ± 0.9 ^a^	27.2 ± 1.1 ^b^	36.7 ± 1.6 ^c^	38.2 ± 0.3 ^c^
*P. gingivalis*	10	13.7 ± 0.5 ^c^	10.7 ± 0.5 ^a^	12 ± 0.2 ^b^	12.7 ± 0.5 ^bc^
	20	18.3 ± 0.5 ^b^	12.8 ± 0.3 ^a^	14 ± 0.7 ^a^	19.1 ± 1 ^b^
	50	28.7 ± 0.2 ^c^	15.5 ± 0.7 ^a^	17.6 ± 0.4 ^b^	28 ± 0.5 ^c^
	100	39.3 ± 0.9 ^b^	18.6 ± 0.4 ^a^	20 ± 0.2 ^a^	39.3 ± 1.1 ^b^
*F. nucleatum*	10	10.7 ± 0.4 ^a^	12.6 ± 0.3 ^b^	14.7 ± 0.4 ^c^	12.3 ± 0.7 ^b^
	20	13.6 ± 0.8 ^a^	18.2 ± 0.5 ^b^	21.9 ± 0.2 ^c^	18 ± 0.2 ^b^
	50	18.4 ± 0.4 ^a^	27.8 ± 1.1 ^b^	34.3 ± 0.9 ^c^	28.3 ± 0.8 ^b^
	100	21 ± 0.1 ^a^	33 ± 0.8 ^b^	43.1 ± 1.6 ^d^	38.8 ± 0.8 ^c^
*P. intermedia*	10	8.9 ± 0.2 ^a^	10.3 ± 0.5 ^b^	12.2 ± 0.3 ^c^	11.3 ± 0.4 ^c^
	20	10.9 ± 0.4 ^a^	13.1 ± 0.3 ^a^	18.9 ± 0.3 ^b^	17.9 ± 0.5 ^b^
	50	17.1 ± 0.3 ^a^	18.9 ± 1 ^a^	31.6 ± 1.3 ^c^	23.7 ± 1 ^b^
	100	22.8 ± 0.6 ^a^	23.2 ± 0.3 ^a^	39.2 ± 0.4 ^c^	28.6 ± 0.9 ^b^
*P. micra*	10	9 ± 0.7 ^a^	10.5 ± 0.4 ^b^	12.5 ± 0.4 ^c^	11.5 ± 0.3 ^bc^
	20	13.2 ± 0.3 ^a^	15 ± 0.5 ^b^	19 ± 0.5 ^c^	18 ± 0.5 ^c^
	50	19.2 ± 0.5 ^a^	21.5 ± 0.6 ^b^	30.5 ± 0.6 ^d^	26 ± 0.3 ^c^
	100	16.2 ± 5.2 ^a^	28.7 ± 0.9 ^b^	40.5 ± 0.6 ^c^	28.2 ± 6.1 ^b^
*S. aureus* ATCC 12600 (Ref)	10	9.2 ± 0.3 ^a^	11.6 ± 0.4 ^bc^	12.3 ± 0.4 ^c^	10.9 ± 0.2 ^b^
	20	13.9 ± 0.5 ^a^	13.5 ± 0.3 ^a^	18.2 ± 0.6 ^b^	14.2 ± 0.3 ^a^
	50	16.9 ± 0.1 ^a^	16.5 ± 0.3 ^a^	21.2 ± 1 ^c^	19.5 ± 0.3 ^b^
	100	18.3 ± 0.3 ^a^	20 ± 0.7 ^a^	26.6 ± 2.7 ^b^	26.8 ± 0.6 ^b^

Different letters in a row indicate significant differences among the various extracts for similar disk content (Kruskal–Wallis with Tukey’s HSD, *p* < 0.05).

**Table 6 microorganisms-12-00060-t006:** Spearman rank correlation coefficients (SRCC) between the antibacterial activities of various Hypericum perforatum extracts (disk diffusion) and disk contents (10, 20, 50, and 100%).

Pathogen	Aqueous	Ethanolic	Aq/Eth Mix	Enzymatic
*S. aureus* MRSA/VRSA	0.972 **	0.907 **	0.842 **	0.972 **
MRSA (raw milk)	0.972 **	0.973 **	0.972 **	0.972 **
MRSA (raw poultry)	0.972 **	0.972 **	0.972 **	0.972 **
*S. mutans*	0.972 **	0.928 **	0.972 **	0.972 **
*S. salivarius*	0.972 **	0.972 **	0.972 **	0.972 **
*P. gingivalis*	0.972 **	0.972 **	0.972 **	0.972 **
*F. nucleatum*	0.972 **	0.972 **	0.972 **	0.972 **
*P. intermedia*	0.972 **	0.972 **	0.972 **	0.972 **
*P. micra*	0.712 *	0.972 **	0.972 **	0.842 **
*S. aureus* ATCC 12600 (Ref)	0.972 **	0.972 **	0.972 **	0.972 **

** p* < 0.05, ** *p* < 0.01 (Spearman rank correlation coefficient, *n* = 12 in all cases).

**Table 7 microorganisms-12-00060-t007:** Antibacterial activities (disk diffusion) of various concentrations (10, 20, 50 and 100%) of aqueous (A), 40% ethanolic (E40), 60% ethanolic (E60), and enzymatic (Enz) extracts from *R. damascene* against various pathogens and a reference strain.

Pathogen	Disk Content (%)	A(mm)	E40 (mm)	E60 (mm)	Enz (mm)
*S. aureus* MRSA/VRSA	10	6.7 ± 0.6 ^a^	10 ± 0.7 ^b^	10.5 ± 0.5 ^b^	6 ± 0 ^a^
	20	11.3 ± 0.4 ^b^	12.6 ± 0.2 ^c^	15 ± 0.4 ^d^	8.5 ± 0.6 ^a^
	50	13.4 ± 0.4 ^b^	17.5 ± 0.5 ^c^	18.9 ± 0.5 ^d^	12.2 ± 0.3 ^a^
	100	17.6 ± 0.3 ^a^	18.5 ± 0.3 ^a^	22.8 ± 2 ^b^	17.9 ± 0.4 ^a^
MRSA (raw milk)	10	7.6 ± 0.6 ^a^	12.5 ± 0.4 ^b^	12.6 ± 0.3 ^b^	7.4 ± 0.5 ^a^
	20	11.9 ± 0.7 ^a^	15 ± 0.2 ^b^	18.6 ± 0.3 ^c^	12 ± 0.4 ^a^
	50	14.8 ± 0.5 ^a^	18 ± 0.3 ^b^	23.1 ± 0.8 ^c^	14.6 ± 0.2 ^a^
	100	17.6 ± 0.3 ^a^	24.3 ± 0.4 ^b^	28.7 ± 0.6 ^c^	17.4 ± 0.4 ^a^
MRSA (raw poultry)	10	6.4 ± 0.4 ^a^	9.1 ± 0.7 ^c^	11.4 ± 0.7 ^d^	7.8 ± 0.5 ^b^
	20	10.6 ± 0.1 ^a^	10.6 ± 0.2 ^a^	16.2 ± 0.6 ^b^	10.7 ± 0.4 ^a^
	50	14.9 ± 0.2 ^a^	16.2 ± 0.3 ^b^	20.6 ± 0.5 ^c^	15.2 ± 0.4 ^a^
	100	17.4 ± 0.4 ^a^	18.3 ± 0.6 ^a^	29.1 ± 0.8 ^b^	17.7 ± 0.3 ^a^
*S. mutans*	10	10.8 ± 0.8 ^c^	8.6 ± 0.3 ^a^	9.8 ± 0.4 ^b^	12.1 ± 0.5 ^d^
	20	16.4 ± 0.4 ^c^	10.7 ± 0.3 ^a^	11.9 ± 0.5 ^b^	17.1 ± 0.2 ^c^
	50	20.7 ± 0.8 ^c^	15.2 ± 0.3 ^a^	16.7 ± 0.5 ^b^	21.5 ± 0.7 ^c^
	100	29.3 ± 0.9 ^c^	16.7 ± 0.2 ^a^	18.5 ± 0.5 ^b^	30.8 ± 0.6 ^d^
*S. salivarius*	10	8.4 ± 0.8 ^a^	10.8 ± 0.6 ^b^	11.7 ± 0.4 ^b^	9.3 ± 0.3 ^a^
	20	12.9 ± 0.6 ^a^	13.1 ± 0.8 ^a^	17.4 ± 0.5 ^b^	12.8 ± 0.5 ^a^
	50	16.4 ± 0.5 ^a^	16.3 ± 0.5 ^a^	22.2 ± 0.7 ^c^	17.8 ± 0.3 ^b^
	100	18.5 ± 0.6 ^a^	18.5 ± 0.5 ^a^	27.5 ± 5.5 ^b^	18.5 ± 0.4 ^a^
*P. gingivalis*	10	11.2 ± 0.8 ^c^	7.6 ± 0.5 ^a^	10.2 ± 0.2 ^b^	12.2 ± 0.6 ^c^
	20	15.2 ± 0.5 ^c^	11.9 ± 0.3 ^a^	14 ± 0.7 ^b^	18.2 ± 0.3 ^d^
	50	20.6 ± 0.5 ^c^	14.1 ± 0.3 ^a^	17 ± 0.2 ^b^	28.1 ± 1 ^d^
	100	32.3 ± 1.9 ^b^	17.5 ± 0.3 ^a^	18.6 ± 0.4 ^a^	33.8 ± 1.2 ^b^
*F. nucleatum*	10	6.1 ± 0.1 ^a^	9.8 ± 0.4 ^b^	11.9 ± 0.6 ^c^	9.8 ± 0.4 ^b^
	20	9.3 ± 0.6 ^a^	10.4 ± 0.4 ^b^	17.2 ± 0.2 ^c^	16.7 ± 0.4 ^c^
	50	12.4 ± 0.6 ^a^	13.9 ± 0.2 ^a^	24.1 ± 1.4 ^c^	19.6 ± 0.8 ^b^
	100	16 ± 0.2 ^a^	16.8 ± 0.2 ^a^	28.9 ± 0.9 ^b^	30.3 ± 0.8 ^c^
*P. intermedia*	10	9.6 ± 0.2 ^b^	8.3 ± 0.6 ^a^	11.7 ± 0.5 ^c^	9 ± 0.1 ^ab^
	20	13.7 ± 0.2 ^b^	12 ± 0.4 ^a^	19 ± 0.4 ^c^	12.5 ± 0.7 ^a^
	50	19 ± 0.2 ^b^	17.3 ± 0.4 ^a^	28.4 ± 0.6 ^c^	16.7 ± 0.4 ^a^
	100	24.5 ± 0.5 ^c^	19.1 ± 0.8 ^b^	31.7 ± 0.6 ^d^	17.9 ± 0.1 ^a^
*P. micra*	10	9.5 ± 0.6 ^a^	10.6 ± 0.6 ^b^	11 ± 0.5 ^bc^	12 ± 0.5 ^c^
	20	12.4 ± 0.4 ^a^	15.3 ± 0.5 ^b^	16.2 ± 0.3 ^b^	19.3 ± 0.5 ^c^
	50	18.1 ± 1 ^a^	21.6 ± 1.2 ^b^	20.8 ± 0.5 ^b^	29.3 ± 0.5 ^c^
	100	21.2 ± 0.9 ^a^	27.8 ± 1.1 ^b^	30.9 ± 0.3 ^c^	35.4 ± 1.1 ^d^
*S. aureus* ATCC 12600 (Ref)	10	9.4 ± 0.5 ^a^	10.9 ± 1.1 ^b^	10.5 ± 0.6 ^ab^	11.1 ± 0.1 ^c^
	20	12.9 ± 0.5 ^a^	14.9 ± 0.1 ^b^	18.1 ± 0.6 ^c^	18 ± 0.2 ^c^
	50	17.5 ± 0.6 ^a^	18.3 ± 1 ^a^	26.7 ± 0.7 ^c^	22.2 ± 1 ^b^
	100	19.8 ± 0.4 ^a^	21 ± 0.4 ^a^	30.7 ± 0.4 ^b^	31.8 ± 2 ^b^

Different letters in a row indicate significant differences among the various extracts with similar disk content (ANOVA with Tukey’s HSD, *p* < 0.05).

**Table 8 microorganisms-12-00060-t008:** Spearman Rank Correlation Coefficients (SRCC) between the antibacterial activities of various *Rosa damascene* extracts (disk diffusion) and disk contents (10, 20, 50, and 100%).

Pathogen	Aqueous	Ethanolic	Aq/Eth Mix	Enzymatic
*S. aureus* MRSA/VRSA	0.972 **	0.972 **	0.950 **	0.865 **
MRSA (raw milk)	0.885 **	0.897 **	0.854 **	0.885 **
MRSA (raw poultry)	0.973 **	0.973 **	0.919 **	0.973 **
*S. mutans*	0.972 **	0.973 **	0.972 **	0.972 **
*S. salivarius*	0.973 **	0.972 **	0.972 **	0.972 **
*P. gingivalis*	0.973 **	0.972 **	0.928 **	0.972 **
*F. nucleatum*	0.972 **	0.972 **	0.972 **	0.972 **
*P. intermedia*	0.972 **	0.972 **	0.972 **	0.972 **
*P. micra*	0.901 **	0.972 **	0.972 **	0.972 **
*S. aureus* ATCC 12600 (Ref)	0.972 **	0.972 **	0.978 **	0.972 **

** *p* < 0.01 (Spearman rank correlation coefficient, *n* = 12 in all cases).

**Table 9 microorganisms-12-00060-t009:** Minimum Inhibitory Concentrations (MIC) and Minimum Bactericidal Concentrations (MBC) of the various extracts (A—Aqueous; E40 and E60—Ethanolic 40% and 60%; and Enz—Enzymatic) from *H. perforatum* against pathogens. Mean values (mg/mL) from three repetitions.

	Minimum Inhibitory Concentration (mg/mL)				Minimum Bactericidal Concentration (mg/mL)			
Pathogen	A	E40	E60	Enz	A	E40	E60	Enz
*S. aureus* MRSA/VRSA	3.1 ± 0 ^4^	1.6 ± 0 ^3^	0.8 ± 0 ^2^	0.4 ± 0 ^1^	4.2 ± 1.8 ^c^	6.3 ± 0 ^d^	1.6 ± 0 ^b^	0.8 ± 0 ^a^
MRSA (raw milk)	6.3 ± 0 ^3^	6.3 ± 0 ^3^	3.1 ± 0 ^2^	0.8 ± 0 ^1^	6.3 ± 0 ^b^	6.3 ± 0 ^b^	6.3 ± 0 ^b^	0.8 ± 0 ^a^
MRSA (raw poultry)	0.8 ± 0 ^2^	1.6 ± 0 ^3^	0.8 ± 0 ^2^	0.4 ± 0 ^1^	0.8 ± 0 ^b^	3.1 ± 0 ^c^	0.8 ± 0 ^b^	0.4 ± 0 ^a^
*S. mutans*	3.1 ± 0 ^2^	6.3 ± 0 ^3^	3.1 ± 0 ^2^	0.4 ± 0 ^1^	6.3 ± 0 ^b^	6.3 ± 0 ^b^	6.3 ± 0 ^b^	0.4 ± 0 ^a^
*S. salivarius*	3.1 ± 0 ^4^	1.6 ± 0 ^3^	0.8 ± 0 ^2^	0.4 ± 0 ^1^	6.3 ± 0 ^d^	3.1 ± 0 ^c^	0.8 ± 0 ^b^	0.4 ± 0 ^a^
*P. gingivalis*	0.4 ± 0 ^1^	6.3 ± 0 ^2^	6.3 ± 0 ^2^	0.4 ± 0 ^1^	0.8 ± 0 ^b^	12.5 ± 0 ^d^	10.4 ± 3.6 ^c^	0.4 ± 0 ^a^
*F. nucleatum*	3.1 ± 0 ^3^	1.6 ± 0 ^2^	0.4 ± 0 ^1^	0.4 ± 0 ^1^	6.3 ± 0 ^d^	3.1 ± 0 ^c^	0.7 ± 0.2 ^b^	0.4 ± 0 ^a^
*P. intermedia*	6.3 ± 0 ^4^	3.1 ± 0 ^3^	0.8 ± 0 ^1^	1.6 ± 0 ^2^	12.5 ± 0 ^d^	8.3 ± 3.6 ^c^	3.1 ± 0 ^b^	1.6 ± 0 ^a^
*P. micra*	3.1 ± 0 ^3^	3.1 ± 0 ^3^	0.8 ± 0 ^1^	1.6 ± 0 ^2^	6.3 ± 0 ^b^	6.3 ± 0 ^b^	1.6 ± 0 ^a^	1.6 ± 0 ^a^
*S. aureus* ATCC 12600 (Ref)	3.1 ± 0 ^2^	3.1 ± 0 ^2^	1.6 ± 0 ^1^	1.6 ± 0 ^1^	6.3 ± 0 ^c^	6.3 ± 0 ^c^	3.1 ± 0 ^b^	3.1 ± 0 ^a^

Different superscript numbers (for MIC) in a row indicate significant differences in MIC among the various extracts. Different superscript letters in a row indicate significant differences in MBC among the various extracts (ANOVA with Tukey’s HSD, *p* < 0.05).

**Table 10 microorganisms-12-00060-t010:** Minimum Inhibitory Concentrations (MIC) and Minimum Bactericidal Concentrations (MBC) of the various extracts (A—Aqueous; E40 and E60—Ethanolic 40% and 60%; and Enz—Enzymatic) from *R. damascene,* against pathogens. Mean values (mg/mL) from three repetitions.

	Minimum Inhibitory Concentration (mg/mL)				Minimum Bactericidal Concentration (mg/mL)			
Pathogen	A	E40	E60	Enz	A	E40	E60	Enz
*S. aureus* MRSA/VRSA	2.6 ± 0.9 ^1^	12.5 ± 0	6.3 ± 0	2.1 ± 0.9 ^1^	5.2 ± 1.8 ^a^	25 ± 0 ^b^	25 ± 0 ^b^	6.3 ± 0 ^a^
MRSA (raw milk)	41.7 ± 14.4 ^4^	25 ± 0 ^3^	5.2 ± 1.8 ^1^	12.5 ± 02 ^2^	70 ± 52 ^d^	50 ± 0 ^c^	6.3 ± 0 ^a^	25 ± 0 ^b^
MRSA (raw poultry)	16.7 ± 7.2 ^2^	12.5 ± 0 ^1^	12.5 ± 0 ^1^	25 ± 0 ^3^	25 ± 0 ^c^	6.3 ± 0 ^a^	12.5 ± 0 ^b^	50 ± 0 ^d^
*S. mutans*	12.5 ± 0 ^3^	8.3 ± 3.6 ^2^	3.1 ± 0 ^1^	3.1 ± 0 ^1^	12.5 ± 0 ^c^	6.3 ± 0 ^b^	6.3 ± 0 ^b^	3.1 ± 0 ^a^
*S. salivarius*	25 ± 0 ^4^	12.5 ± 0 ^3^	3.1 ± 0 ^1^	6.3 ± 0 ^2^	50 ± 0 ^d^	25 ± 0 ^c^	3.1 ± 0 ^a^	12.5 ± 0 ^b^
*P. gingivalis*	3.1 ± 0 ^1^	12.5 ± 0 ^2^	12.5 ± 0 ^2^	3.1 ± 0 ^1^	6.3 ± 0 ^b^	16.7 ± 7 ^a^	12.5 ± 0 ^c^	4.2 ± 1.8 ^a^
*F. nucleatum*	0.8 ± 0 ^2^	1.6 ± 0 ^3^	0.4 ± 0 ^1^	3.1 ± 0 ^4^	1.6 ± 0 ^b^	1.6 ± 0 ^b^	0.8 ± 0 ^a^	3.1 ± 0 ^c^
*P. intermedia*	0.8 ± 0 ^2^	1.6 ± 0 ^3^	0.4 ± 0 ^1^	3.1 ± 0 ^4^	3.1 ± 0 ^b^	3.1 ± 0 ^b^	0.8 ± 0 ^a^	6.3 ± 0 ^c^
*P. micra*	25 ± 0 ^3^	12.5 ± 0 ^2^	12.5 ± 0 ^2^	0.4 ± 0 ^1^	50 ± 0 ^d^	25 ± 0 ^c^	16.7 ± 7.2 ^b^	0.7 ± 0.2 ^a^
*S. aureus* ATCC 12600 (Ref)	3.1 ± 0 ^2^	3.1 ± 0 ^2^	2.1 ± 0.9 ^2^	0.8 ± 0 ^1^	12.5 ± 0 ^d^	6.3 ± 0 ^c^	3.1 ± 0 ^b^	1.6 ± 0 ^a^

Different superscript numbers (for MIC) in a row indicate significant differences in MIC among the various extracts. Different superscript letters in a row indicate significant differences in MBC among the various extracts (ANOVA with Tukey’s HSD, *p* < 0.05).

**Table 11 microorganisms-12-00060-t011:** Mean antibacterial activities (disk diffusion in mm) of aqueous, ethanolic, mixed, and enzymatic extracts from *H. perforatum* and *R. damascene* against pathogens and a reference strain.

Extract/Herbal		N	Mean ± SD
Aqueous	*Hypericum perforatum*	120	17.43 ± 7.03 ^b^
	*Rosa damascene*	120	14.76 ± 6.02 ^a^
Ethanolic	*Hypericum perforatum*	120	17.55 ± 6.37 ^b^
	*Rosa damascene*	120	14.62 ± 4.59 ^a^
Aqueous/Ethanolic Mix	*Hypericum perforatum*	120	21.64 ± 8.81 ^c^
	*Rosa damascene*	120	16.58 ± 5.01 ^a^
Enzymatic	*Hypericum perforatum*	120	23.08 ± 9.73 ^b^
	*Rosa damascene*	120	16.66 ± 6.64 ^a^

Different superscript letters indicate significant differences in mean disk diffusion results between the various plant species for each type of extract (Kruskal–Wallis with Tukey’s HSD, *p* < 0.05).

**Table 12 microorganisms-12-00060-t012:** Spearman rank correlation coefficients between MIC concentrations (×8, ×4, ×2, ×1, ×0.5) and antibiofilm effectiveness (2 herbal species, 9 pathogens, 1 reference strain).

Antibiofilm Agent	Corr. Coefficient, (2-Tailed Sig.), Sample N
Aqueous extract	0.155, (<0.05), 198
Ethanolic extract	0.111 (>0.05), 199
Aqueous/Ethanolic Mix extract	0.014, (>0.05), 197
Enzymatic extract	−0.091, (>0.05, 197
Clidamycin	0.761, (<0.001), 96
Ciprofloxacin	0.732, (>0.05), 92
Gentamycin	0.712, (<0.001), 92
Imipenem	0.778, (<0.001), 196
Vancomycin	0.638, (<0.001), 100
Metronidazol	0.775, (<0.001), 100
Amoxicillin/Clavulanic Acid	0.812, (<0.001), 100

**Table 13 microorganisms-12-00060-t013:** Time-kill results (mean log cfu/mL ± St. Dev) of aqueous and enzymatic extracts from *R. damascene* against 9 pathogens and a reference strain. The extracts were tested at typical concentrations of ×1, ×2, ×4 and ×8 times the respective MBC as evaluated using the broth micro-dilution method. Counts were made at 0, 2, 4, 8, 12, and 24 h of the incubation period.

Pathogen	MBC	Aqueous Extract						Enzymatic Extract						Control (No Extract)					
		0 h	2 h	4 h	8 h	12 h	24 h	0 h	2 h	4 h	8 h	12 h	24 h	0 h	2 h	4 h	8 h	12 h	24 h
*S. aureus* MRSA/VRSA	8	6.27 ± 0.21	5.43 ± 1.02	4.83 ± 0.91	3.97 ± 0.21	3.83 ± 0.8	0 ± 0	5.77 ± 1.29	4.57 ± 1.52	4.57 ± 0.42	2.23 ± 1.4	0 ± 0	0 ± 0	6.27 ± 1.42	5.9 ± 0.69	5.97 ± 0.31	11.77 ± 0.47	12.37 ± 0.21	13.1 ± 1.21
	4	6.27 ± 0.29	5.47 ± 1.18	5 ± 0.2	4.67 ± 1.59	3.97 ± 1.02	0 ± 0	5.83 ± 0.21	4.87 ± 0.6	4.63 ± 0.9	2.4 ± 0.44	0 ± 0	0 ± 0						
	2	6 ± 0.69	6.17 ± 0.4	6.2 ± 1.3	5.7 ± 1.21	4.93 ± 0.67	1.03 ± 0.74	5.83 ± 0.31	4.97 ± 0.47	4.83 ± 0.38	2.93 ± 0.21	0.5 ± 0	0 ± 0						
	1	6.23 ± 1.06	6.23 ± 1.51	6.57 ± 1.1	6.03 ± 0.51	5.83 ± 1.4	2.97 ± 1.03	5.83 ± 0.78	5.63 ± 0.6	5.63 ± 1.38	3.37 ± 0.68	1.8 ± 0.61	0 ± 0						
MRSA (milk)	8	5.83 ± 0.68	5.43 ± 0.81	4.9 ± 0.52	3.77 ± 0.81	2.5 ± 1.01	0 ± 0	5.83 ± 0.21	5.3 ± 1.51	2.93 ± 0.7	2.47 ± 0.78	0 ± 0	0 ± 0	6.13 ± 0.5	6.03 ± 0.42	6.03 ± 0.21	6.9 ± 0.3	10.03 ± 1.23	12.63 ± 0.99
	4	5.93 ± 1.4	5.93 ± 0.68	5.07 ± 0.29	3.93 ± 0.31	3.8 ± 0.52	0 ± 0	5.83 ± 1.24	5.37 ± 0.12	3.37 ± 0.29	2.8 ± 1.4	0 ± 0	0 ± 0						
	2	5.97 ± 0.49	5.93 ± 0.9	5.93 ± 0.61	5.07 ± 1.38	4.33 ± 0.7	0.83 ± 0.72	5.83 ± 0.25	5.73 ± 0.92	4.57 ± 1.12	3.93 ± 0.29	0 ± 0	0 ± 0						
	1	6.1 ± 0.26	5.97 ± 0.25	5.97 ± 0.58	5.73 ± 1.39	5.5 ± 0.5	2.03 ± 1.39	5.93 ± 0.9	5.77 ± 1.11	5 ± 0.17	4.77 ± 1.5	4.03 ± 0.5	0 ± 0						
MRSA (poultry)	8	5.97 ± 0.7	5.97 ± 1.32	5.7 ± 0.56	4.4 ± 1.14	2.93 ± 1.31	0 ± 0	5.83 ± 0.59	4.93 ± 0.6	4.1 ± 0.1	2.93 ± 1.1	0 ± 0	0 ± 0	6.43 ± 0.55	6 ± 0.52	6.03 ± 0.9	7.07 ± 0.81	10.8 ± 0.36	12.83 ± 1.19
	4	5.97 ± 1.3	6 ± 0.52	6 ± 0.5	5 ± 0.2	3.7 ± 0.61	0.83 ± 0.49	5.87 ± 0.4	5.07 ± 0.32	4.97 ± 1.2	3.2 ± 1.3	0 ± 0	0 ± 0						
	2	5.97 ± 0.59	6.2 ± 0.7	6.2 ± 0.1	5.5 ± 0.7	4.57 ± 1.29	1.53 ± 0.21	5.87 ± 0.21	5.77 ± 0.72	5 ± 1.22	3.93 ± 1.4	2.3 ± 1.51	0 ± 0						
	1	5.97 ± 0.32	6.2 ± 1.22	6.73 ± 0.78	5.9 ± 0.62	5.8 ± 0.17	2.8 ± 0.87	5.93 ± 1.4	5.87 ± 0.4	4.83 ± 0.61	3.97 ± 1.27	3.7 ± 1.31	0 ± 0						
*S. mutans*	8	5.8 ± 0.6	3.97 ± 1.04	2.07 ± 0.93	0 ± 0	0 ± 0	0 ± 0	5.93 ± 0.15	4.97 ± 1.11	4.13 ± 0.12	2.03 ± 0.93	0 ± 0	0 ± 0	5.93 ± 0.59	6 ± 0.52	6.03 ± 1.1	6.87 ± 0.99	8.83 ± 1.5	11.8 ± 0.36
	4	5.83 ± 0.86	3.97 ± 0.51	2.37 ± 0.21	0 ± 0	0 ± 0	0 ± 0	5.9 ± 1.11	5.07 ± 0.31	4.23 ± 0.12	2.07 ± 0.31	0 ± 0	0 ± 0						
	2	5.83 ± 1.51	4.8 ± 0.2	2.8 ± 0.44	0.87 ± 0.8	0 ± 0	0 ± 0	5.9 ± 0.2	5.83 ± 0.7	4.97 ± 0.12	3.4 ± 1.39	0 ± 0	0 ± 0						
	1	5.97 ± 0.31	4.97 ± 1.27	4.03 ± 0.12	3.07 ± 0.7	0 ± 0	0 ± 0	5.97 ± 1.1	6.1 ± 0.5	6.03 ± 0.06	5.7 ± 1.3	3 ± 0.1	1.1 ± 0.1						
*S. salivarius*	8	5.8 ± 0.72	2.93 ± 0.59	1 ± 1.05	0 ± 0	0 ± 0	0 ± 0	5.83 ± 1.18	6 ± 0.4	4.9 ± 0.53	2.07 ± 1.31	0 ± 0	0 ± 0	5.87 ± 0.61	6.23 ± 0.38	6.23 ± 0.9	7 ± 1.48	8.6 ± 0.7	11.03 ± 0.5
	4	5.67 ± 1.21	3.07 ± 0.31	1.03 ± 0.31	0.23 ± 0.95	0 ± 0	0 ± 0	5.9 ± 0.3	5.97 ± 1.07	4.93 ± 0.4	2.07 ± 0.9	0 ± 0	0 ± 0						
	2	5.87 ± 0.21	4.87 ± 0.5	3.2 ± 1.31	1.4 ± 0.61	0 ± 0	0 ± 0	5.93 ± 1.48	6 ± 0.7	5.03 ± 0.71	3.87 ± 1.27	2 ± 0.1	0 ± 0						
	1	6.07 ± 0.9	5.3 ± 1.31	4 ± 0.1	2.93 ± 1.1	0 ± 0	0 ± 0	5.93 ± 0.67	6 ± 0.1	6.07 ± 0.38	4.83 ± 0.5	2.53 ± 0.15	0.9 ± 0.2						
*P. gingivitis*	8	5.87 ± 0.12	3.47 ± 1.22	1 ± 0.2	0.43 ± 0.12	0 ± 0	0 ± 0	5.8 ± 1.11	3.03 ± 0.06	1.03 ± 0.49	−0.03 ± 0.35	0 ± 0	0 ± 0	5.87 ± 1.44	5.97 ± 0.21	5.97 ± 1.52	7.07 ± 1.11	8.97 ± 1.4	11.47 ± 1.5
	4	5.83 ± 0.12	5.13 ± 0.51	1.97 ± 0.95	1.1 ± 0.7	0 ± 0	0 ± 0	5.87 ± 1.1	4.47 ± 1.31	3.1 ± 1.05	1.8 ± 0.78	0 ± 0	0 ± 0						
	2	5.87 ± 0.71	5.93 ± 1.01	5.2 ± 0.1	4.53 ± 0.42	0 ± 0	0 ± 0	5.87 ± 0.61	5.87 ± 0.21	5.73 ± 1.07	1.97 ± 1.21	0 ± 0	0 ± 0						
	1	5.9 ± 0.1	6 ± 1.21	5.73 ± 1.16	5.03 ± 1.42	2.4 ± 1.13	0.57 ± 0.4	5.87 ± 1.52	5.83 ± 0.12	5.73 ± 0.57	2.97 ± 1.12	1.07 ± 0.93	0 ± 0						
*F. nucleatum*	8	5.87 ± 0.81	3.97 ± 0.49	1.93 ± 0.32	0.23 ± 0.87	0 ± 0	0 ± 0	5.8 ± 0.26	4.07 ± 1.1	1.9 ± 1.5	0.03 ± 1.05	0 ± 0	0 ± 0	5.97 ± 0.31	6.83 ± 0.9	6.8 ± 0.2	8.83 ± 0.5	9.47 ± 0.67	10.93 ± 0.32
	4	5.87 ± 1.15	4.97 ± 1.02	2.53 ± 0.38	1.5 ± 1.31	0 ± 0	0 ± 0	5.83 ± 1.08	4.1 ± 0.75	2.83 ± 0.47	1.03 ± 0.67	0 ± 0	0 ± 0						
	2	5.83 ± 0.4	5.17 ± 1.17	3.37 ± 0.9	2.5 ± 0.79	0 ± 0	0 ± 0	5.87 ± 0.31	4.83 ± 0.32	3.03 ± 0.29	1.93 ± 0.42	0 ± 0	0 ± 0						
	1	5.93 ± 0.85	5.97 ± 0.47	3.93 ± 0.32	2.93 ± 0.5	1.5 ± 0.1	0 ± 0	5.87 ± 0.21	5.87 ± 0.96	4.8 ± 0.1	2.83 ± 0.06	0 ± 0	0 ± 0						
*P. intermedia*	8	4.8 ± 0.1	3.57 ± 0.61	1.77 ± 1.54	0.1 ± 0.1	0 ± 0	0 ± 0	5.8 ± 0.78	5.3 ± 1.2	3.83 ± 0.91	1.97 ± 0.5	0 ± 0	0 ± 0	5.87 ± 0.83	6 ± 0.1	6 ± 0.2	7.93 ± 1.08	8.77 ± 0.31	9.97 ± 1.32
	4	5.8 ± 0.1	4.77 ± 1.51	2.4 ± 0.2	0.34 ± 0.05	0 ± 0	0 ± 0	5.77 ± 0.6	5.37 ± 0.64	3.93 ± 0.21	2.07 ± 0.5	0 ± 0	0 ± 0						
	2	5.97 ± 0.31	4.83 ± 0.32	3.07 ± 1.3	2.03 ± 1.29	0 ± 0	0 ± 0	5.83 ± 1.16	5.5 ± 0.9	4.37 ± 0.9	2.87 ± 1	0 ± 0	0 ± 0						
	1	5.93 ± 0.9	5.37 ± 1.12	3.17 ± 0.29	3.13 ± 1.3	1 ± 0.17	0.07 ± 0.01	5.83 ± 0.38	5.5 ± 1.28	4.9 ± 0.61	3.97 ± 1	1.7 ± 0.2	0 ± 0						
*P. micra*	8	5.63 ± 0.12	4.8 ± 1.11	0.93 ± 0.4	0.4 ± 0.53	0 ± 0	0 ± 0	5.83 ± 1.29	5.57 ± 0.21	3.97 ± 1.1	3.8 ± 0.8	0 ± 0	0 ± 0	5.97 ± 1.33	7 ± 0.7	6.97 ± 1.1	8.87 ± 1.21	8.97 ± 0.4	10.8 ± 0.69
	4	5.67 ± 1.5	4.97 ± 0.32	2.43 ± 1	0.97 ± 0.47	0 ± 0	0 ± 0	5.87 ± 0.59	5.53 ± 1.06	4.67 ± 0.67	3.97 ± 0.7	0 ± 0	0 ± 0						
	2	5.73 ± 0.76	4.97 ± 0.29	3.07 ± 0.23	1.97 ± 0.9	0 ± 0	0 ± 0	5.9 ± 0.6	5.87 ± 0.4	4.73 ± 1.33	4.03 ± 1.4	0 ± 0	0 ± 0						
	1	5.93 ± 0.42	5.77 ± 0.4	4.03 ± 1.19	2.07 ± 0.6	1.03 ± 0.38	0.08 ± 0.01	5.93 ± 1.18	5.9 ± 1.31	5.83 ± 0.81	5.57 ± 0.38	2.03 ± 0.32	0.9 ± 0.1						
*S. aureus* ATCC 12600 (Ref)	8	5.57 ± 0.32	4.97 ± 0.9	3.6 ± 0.98	2 ± 0.98	0.8 ± 0.1	0 ± 0	0 ± 0	4.4 ± 0.2	4 ± 1.61	2.8 ± 1.1	0 ± 0	0 ± 0	5.93 ± 0.59	6.07 ± 0.78	6.07 ± 0.06	7.77 ± 0.23	8.07 ± 1.31	8.9 ± 0.78
	4	5.57 ± 1.43	5.03 ± 0.29	4.1 ± 0.2	1.97 ± 0.4	0.87 ± 0.21	0 ± 0	0 ± 0	4.67 ± 0.12	4.07 ± 0.42	2.5 ± 0.1	0 ± 0	0 ± 0						
	2	5.63 ± 0.76	5.63 ± 1.4	4.83 ± 0.38	3.07 ± 1.3	1 ± 0.7	0 ± 0	0 ± 0	5.03 ± 1.52	4.53 ± 1	3.07 ± 0.49	0.3 ± 0.1	0 ± 0						
	1	5.97 ± 1.19	6 ± 0.26	5.6 ± 0.5	5 ± 0.1	2.77 ± 1.16	0.13 ± 0.01	5.93 ± 0.71	5.67 ± 1.1	5.63 ± 1.44	3.37 ± 1.19	1.03 ± 0.12	0 ± 0						

**Table 14 microorganisms-12-00060-t014:** Bactericidal effectiveness (in hours) of the aqueous and enzymatic extracts from *R. damascene* with respect to the MBCs estimated using time-kill kinetics. A decrease in the initial bacteria population by at least 3 log CFU/mL was considered the bactericidal effect of the studied concentration.

Pathogen	Concentration (×MBC)	Aqueous Extract Killing-Time (h) (Hours)	Enzymatic Extract Killing-Time (h) (Hours)
*S. aureus* MRSA/VRSA	8	24	8
	4	24	8
	2	24	8
	1	24	12
MRSA (raw milk)	8	12	8
	4	24	8
	2	24	12
	1	24	24
MRSA (raw poultry)	8	12	12
	4	24	12
	2	24	12
	1	24	24
*S. mutans*	8	4	8
	4	4	8
	2	4	12
	1	12	24
*S. salivarius*	8	2	8
	4	4	8
	2	4	12
	1	8	12
*F. nucleatum*	8	4	4
	4	4	8
	2	12	8
	1	12	12
*P. gingivitis*	8	4	8
	4	4	4
	2	8	8
	1	8	8
*P. intermedia*	8	4	8
	4	4	8
	2	8	8
	1	12	12
*P. micra*	8	4	8
	4	4	12
	2	8	12
	1	8	12
*S. aureus* ATCC 12600	8	8	12
	4	8	12
	2	12	12
	1	12	12

## Data Availability

Data are contained within the article.

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
