# Peer review of "The In Vitro Assessment of Antibacterial and Antioxidant Efficacy in Rosa damascena and Hypericum perforatum Extracts against Pathogenic Strains in the Interplay of Dental Caries, Oral Health, and Food Microbiota"

_microorganisms, 2023, doi:10.3390/microorganisms12010060_

Round 1

Reviewer 1 Report

Comments and Suggestions for Authors

The topic of the manuscript    is relevant . The introduction presents the rationale for the study. However, the manuscript  needs to be thoroughly revised and shortened. There is a lot of duplicate data and methods are described incorrectly.

1.The methods of preparation of extracts are written incorrectly. It is not clear which part of the plant was extracted.

The authors wrote: line 144«The collected plant samples were left to dry at room temperature. Once completely dried, samples were separated into different plant parts, i.e., roots, leaf, fruits, bark, and stems. Dried plant materials (for the reses only rose petals, calyces and pollen” what is reses?

 line 154 The aqueous extract of roses used in the present study came from the Women's Co- operative of Kozani. Approximately 10 g of powder plant were dissolved…….  Should write which part of the plant was used for extraction

The same  for Hypericum perforatum.

2. Line 161 For the other plant, aqueous extracts were prepared by immersing plant powder material in sterile distilled water.  Should write which part of the plant was used for extraction

3. Lines 162-163 Plant powder was suspended in 1:2 distilled water at a ratio of plant powder: solvent of 1:2, … What's the ratio? Weight or volume?

4. Line 174 «1 kg of plant........”  Should write which part of the plant was used for extraction

5 .Line 176After incubation at 370 C for 48 h, hydrolysis was interrupted by heating for 10 min and then the solvent was evaporated by means of a rotary evaporatorThis method does not include any stage of filtration? This method needs to be described in detail as it is rarely used.

 6.Line 200 "Fifty milligrams of the extract were dissolved in distilled water". Fifty milligrams of the extract were dissolved in what volume of distilled water?

7.Line 205 “The extract was put in a test tube and mixed with 2 mL of chloroform and then concentrated sulfuric acid (H2SO4) (3 ml) was added to form a layer”. How much extract and in what form (powder, solution) was put into the tube?

8.Line 213  Fifty milligrams of the extract were dissolved in 5 mL of alcohol  …..” What concentration of alcohol was used?

9. Line 228The total flavonoids were expressed as milligrams of gallic acid equivalents (GAE) per g of sample”. Why was gallic acid, which is not a flavonoid, used as a standard for measuring flavonoid levels rather than a flavonoid such as quercetin?

10.Line 231  « ......2,20 - diphenyl-1-picrylhydrazyl should be   ( 2,2-diphenyl-1-picrylhydrazyl

11.Line 238 «The control was prepared as above without any sample”. What does it mean without any sample? The control have to  contain the same volume of solvent as was used for extract.

12.All descried methods should be referenced

13. line 343In order to in order to certify the process wells containing solely the used nutrient medium liquid, the liquid medium with inoculum or 0.2% chlorhexidine (CHX) served as controls. Please correct this sentence

14.Line 430  Table 2 and figure 1  duplicate data

15.Line 457Increased extracts concentrations resulted in higher reducing power in both plants ‘ should write Increased extracts concentrations resulted in higher reducing power these samples.

16.Line 491-494’ Why in the disc diffusion  experiments the concentrations of extract are presented as percentages while in the experiments to determine the MIC and MBC  concentration expressed as mg/ml? This differences  makes it difficult to compare results.

17.Line496. «Similarly, pure enzymatic extract was equally.....» What does purified
extract mean?
In the described method there is not purification stage. 

18.  The data on the antibacterial activity of the extracts are presented as MIC and MBC in tables 9 and 10. Data of disk diffusion experiments are presented in tables 5 and 6.  Is it completely unclear what kind of antibacterial activity is shown in Table 11? It is not clear that  concentrations were used, what  pathogens were used? what "total" means?  These data do not provide any additional information about antimicrobial activity of extracts.

19. Table 12 and figure 2  duplicate data. There are no statistical outliers on bars in figure 2

20.In section “ Antibiofilm effectiveness” is Fig 3  Mean Inhibition zones (± standard deviation) of the various herb extracts and antibiotics against oral and other pathogens’ Description of Fig 3  is not in the text.

21. In addition to the MIC and MBC values, what additional characterisation of the antimicrobial activity of the extracts was provided by the results of the diffusion test in agar?

Discussion

1. It remains unclear why flavonoids that are poorly soluble in water are better extracted with water  than  40% and 60%  in the case of R. damascene . Please explain 

2. Section 4.4  title should be change .Instead of “Biological functions of extracts” should write “Antimicrobial and antioxidant activity of the studied extracts

Comments on the Quality of English Language

Some sentences need to be corrected. I have indicated them above

Author Response

Dear Reviewer 1,

Thank you for your advice and your review of our manuscript in order to upgrade its quality. We are grateful for your comments. We do hope that our revision meets your requirements. Replies to your valuable comments are given below point by point. Changes were highlighted with yellow color in the manuscript.

Point 1: The authors wrote: line 144«The collected plant samples were left to dry at room temperature. Once completely dried, samples were separated into different plant parts, i.e., roots, leaf, fruits, bark, and stems. Dried plant materials (for the reses only rose petals, calyces and pollen” what is reses?

Response 1: You are absolutely right. This is a typographical error. Correction done. Please refer to lines 146-148.

Point 2: line 154 The aqueous extract of roses used in the present study came from the Women's Co- operative of Kozani. Approximately 10 g of powder plant were dissolved…….  Should write which part of the plant was used for extraction. The same  for Hypericum perforatum.

Response 2: Please refer to Lines 146-148.

Point 3. Line 161 For the other plant, aqueous extracts were prepared by immersing plant powder material in sterile distilled water.  Should write which part of the plant was used for extraction

Response 3: Please refer to Lines 146-148.

Point 4. Lines 162-163 Plant powder was suspended in 1:2 distilled water at a ratio of plant powder: solvent of 1:2, … What's the ratio? Weight or volume?

Response 4: Corrected, please refer to Lines 163-164.

Point 5. Line 174 «1 kg of plant........”  Should write which part of the plant was used for extraction.

Response 5: You are absolutely right. Please refer to Lines 175-176.

Point 6 . Line 176 “After incubation at 370 C for 48 h, hydrolysis was interrupted by heating for 10 min and then the solvent was evaporated by means of a rotary evaporator” This method does not include any stage of filtration? This method needs to be described in detail as it is rarely used.

Response 6: Done. Please refer to Lines 175-187.

 Point 7. Line 200 "Fifty milligrams of the extract were dissolved in distilled water". Fifty milligrams of the extract were dissolved in what volume of distilled water?

Response 7: The required data added, please refer to Lines 209-213.

Point 8. Line 205 “The extract was put in a test tube and mixed with 2 mL of chloroform and then concentrated sulfuric acid (H2SO4) (3 ml) was added to form a layer”. How much extract and in what form (powder, solution) was put into the tube?

Response 8: The required data added, please refer to Lines 215-217.

Point 9. Line 213  “Fifty milligrams of the extract were dissolved in 5 mL of alcohol  …..” What concentration of alcohol was used?

Response 9: Corrected, please refer to Lines 221-222.

Point 10. Line 228 “The total flavonoids were expressed as milligrams of gallic acid equivalents (GAE) per g of sample”. Why was gallic acid, which is not a flavonoid, used as a standard for measuring flavonoid levels rather than a flavonoid such as quercetin?

Response 10: Gallic acid is classified as a phenolic acid (https://pubchem.ncbi.nlm.nih.gov/compound/Gallic-Acid). On the other hand, the flavonoids, are a group of natural substances with variable phenolic structures (Panche, A.N., Diwan, A.D., Chandra, S.R., 2016. Flavonoids: an overview. Journal of Nutritional Science 5. https://doi.org/10.1017/jns.2016.41). Nevertheless, in the present study this is a typographical error, because the word gallic acid remained from the previously described method. The right compound is catechin equivalents (CE). Please refer to Lines 241-242.

Point 11. Line 231  « ......2,20 - diphenyl-1-picrylhydrazyl should be   ( 2,2-diphenyl-1-picrylhydrazyl)

Response 11: Corrected

Point 12. Line 238 «The control was prepared as above without any sample”. What does it mean without any sample? The control have to contain the same volume of solvent as was used for extract.

Response 12: That is exactly what we meant: same volume with only the solvent. Furthermore, the amount of 100 μL, which consist of the adding studding extract is considered negligible therefore the lack of this small amount from the total solution that we use as a control we consider that it does not create any problem in the whole process.

Point 13. All descried methods should be referenced.

Response 13: Done.

Point 14. line 343 “In order to in order to certify the process wells containing solely the used nutrient medium liquid, the liquid medium with inoculum or 0.2% chlorhexidine (CHX) served as controls. Please correct this sentence.

Response 14: Connected. Please refer to Lines 355-357.

Point 15. Line 430  Table 2 and figure 1  duplicate data

Response 15: We thank the reviewer for the comment. Same data but with a different statistical analysis. In table 2, total phenolics and total flavonoids are compared between the different extracts for each one of the two plant species separately. In Figure 1, the above variables are compared regardless of the plant. 

Point 16. Line 457 ‘Increased extracts concentrations resulted in higher reducing power in both plants ‘ should write Increased extracts concentrations resulted in higher reducing power these samples.

Response 16: Done, please refer to Line 466.

Point 17. Line 491-494’ Why in the disc diffusion experiments the concentrations of extract are presented as percentages while in the experiments to determine the MIC and MBC concentration expressed as mg/ml? This differences makes it difficult to compare results.

Response 17: We thank the reviewer for the comment. Diffusion experiments can be regarded as screening tests in order to study antimicrobial activities in a wide range of concentrations. While such approach is equally important and often used in published studies to accommodates comparison, in order for us to proceed with killing time experiments, MIC and MBC had to be estimated accurately and in much lower concentrations (below 20%) in which case mg/ml applied. Additionally, it is rather easier for someone to comprehend 0.0975 mg/ml than 0.00975% which was the lowest concentration used in MIC/MBC experiments. In every case, we agree with the reviewer about the difficulty occurred and in order to justify this shifting from percentage to mg/ml the text (in L343-345) was altered as follows: “For an accurate estimation and in much lower concentrations than in agar diffusion experiments, the minimum inhibitory concentration was estimated by employing the microdilution broth method in 96-well microplates [55,59,60].

Point 18. Line 496. «Similarly, pure enzymatic extract was equally.....» What does purified extract mean? In the described method there is not purification stage.

Response 18: Corrected, the word pure deleted.

Point 19. The data on the antibacterial activity of the extracts are presented as MIC and MBC in tables 9 and 10. Data of disk diffusion experiments are presented in tables 5 and 6.  Is it completely unclear what kind of antibacterial activity is shown in Table 11? It is not clear that concentrations were used, what  pathogens were used? what "total" means?  These data do not provide any additional information about antimicrobial activity of extracts.

Response 19: Thank you for your comment. Table 11 can be regarded as a summary table summarizing the data from tables 5 and 6.  In table 11 it is more obvious the effectiveness of all H. perforatum extracts in comparison with those of R. damascene as statistical analysis indicates and support the findings described in the text (L551-552). Totals, are the mean values from both plants for each type of extract. To avoid confusion, totals and the last column of the table (significance) had been removed. 

Point 20. Table 12 and figure 2 duplicate data. There are no statistical outliers on bars in figure 2

Response 20. Thank you for the comment. In Figure 2 the results of the statistical comparison are also shown however, table 12 was deleted while error bars have been added in Fig. 2 as recommended.

Point 21. In section “ Antibiofilm effectiveness” is Fig 3  “Mean Inhibition zones (± standard deviation) of the various herb extracts and antibiotics against oral and other pathogens’ Description of Fig 3  is not in the text.

Response 21: “(Figure 3)” in L582 added in the text

Point 22. In addition to the MIC and MBC values, what additional characterisation of the antimicrobial activity of the extracts was provided by the results of the diffusion test in agar?

Response 22: Thank you for your comment. Similar to the response in comment 17, disk diffusion experiments served us as a screening approach regarding the antimicrobial effectiveness and often in literature this is the only method used to support any findings. However, we consider that based on disk diffusion results (i.e. effectiveness in low concentration of the various extracts) we had to procced with a more elaborate analysis such as the MIC/MBC estimation, killing-time experiments etc. Other than that, agar diffusion experiments can contribute further to study variables like the role of the type of the solvent, the diffusion of antimicrobials and aerobic-anaerobic conditions among others.

Discussion

Point 1. It remains unclear why flavonoids that are poorly soluble in water are better extracted with water  than  40% and 60%  in the case of R. damascene.  Please explain.

Response 1: It is true that the free flavonoids are insoluble or dissoluble in water. A possible explanation for our results, is that these flavonoids are perhaps in vivo conjugated with another substance and this complex might be water soluble. Another explanation is that the aqueous extract of the Rosa damascene, usually contains some trace quantities of the essential oil of the plant, which is rich in flavonoids. It is more than obvious that further research is needed.

Point 2. Section 4.4  title should be change .Instead of “Biological functions of extracts” should write “Antimicrobial and antioxidant activity of the studied extracts”

Response 2: Done.

Thank you for your time and effort

The authors

Reviewer 2 Report

Comments and Suggestions for Authors

General Comment:

I would like to commend the authors for their well-written and scientifically sound research paper titled "In Vitro Assessment of Antibacterial and Antioxidant Efficacy in Rosa damascena and Hypericum perforatum Extracts Against Pathogenic Strains in the Interplay of Dental Caries, Oral Health, and Food Microbiota". The research addresses an important area of study linking dental caries, oral health, food microbiota, and the potential antibacterial and antioxidant effects of two plant extracts. Overall, I find this manuscript suitable for publication.

Specific Comments:

1. Title and Abstract:

The title accurately reflects the main focus of the study and effectively captures the essence of the research. The abstract provides a concise summary of the objectives, methods, and findings of the study.

2. Introduction:

The introduction provides a clear background on the interplay of dental caries, oral health, and food microbiota, establishing a context for the research. It effectively highlights the significance of investigating alternative treatment options using natural extracts. I recommend the authors consider adding a brief mention of the current challenges in dental healthcare and the potential role of plant-based therapies, providing a broader perspective.

3. Materials and Methods:

The materials and methods section is comprehensive, allowing for the replication of the study. The description of plant extract preparation, bacterial strains, and experimental setup is detailed enough to ensure reproducibility. However, providing specific information regarding the concentrations and types of antibacterial agents or antioxidants present in the plant extracts would enhance the clarity of the study.

4. Results and Discussion:

The results section presents a thorough analysis of the antibacterial and antioxidant efficacy of the Rosa damascena and Hypericum perforatum extracts against pathogenic strains. The data is well-organized and visually represented, aiding in a clear understanding of the findings. The discussion succinctly interprets the results and relates them to the existing literature. I suggest the authors consider expanding on the potential mechanisms of action underlying the observed antimicrobial and antioxidant activity to provide a more insightful discussion.

5. Conclusion:

The conclusion aptly summarizes the key findings and their implications within the context of the study. However, I would encourage the authors to provide a brief statement on the limitations of the research and propose future directions to further investigate the potential clinical applications of the studied plant extracts.

Comments on the Quality of English Language

Language and Presentation: The manuscript is well-written, and the language throughout is clear and concise. However, a few minor typographical errors were noticed, which can be easily corrected during the proofreading stage. The figures and tables are appropriately labeled and contribute to the overall understanding of the research.

Author Response

REVIEWER 2

General Comment:

I would like to commend the authors for their well-written and scientifically sound research paper titled "In Vitro Assessment of Antibacterial and Antioxidant Efficacy in Rosa damascena and Hypericum perforatum Extracts Against Pathogenic Strains in the Interplay of Dental Caries, Oral Health, and Food Microbiota". The research addresses an important area of study linking dental caries, oral health, food microbiota, and the potential antibacterial and antioxidant effects of two plant extracts. Overall, I find this manuscript suitable for publication.

Specific Comments:

  1. Title and Abstract:

The title accurately reflects the main focus of the study and effectively captures the essence of the research. The abstract provides a concise summary of the objectives, methods, and findings of the study.

  1. Introduction:

The introduction provides a clear background on the interplay of dental caries, oral health, and food microbiota, establishing a context for the research. It effectively highlights the significance of investigating alternative treatment options using natural extracts. I recommend the authors consider adding a brief mention of the current challenges in dental healthcare and the potential role of plant-based therapies, providing a broader perspective.

Response: We added relevant information. Please see lines 113-134.

  1. Materials and Methods:

The materials and methods section is comprehensive, allowing for the replication of the study. The description of plant extract preparation, bacterial strains, and experimental setup is detailed enough to ensure reproducibility. However, providing specific information regarding the concentrations and types of antibacterial agents or antioxidants present in the plant extracts would enhance the clarity of the study.

Response: Thank you very much for your comments. We have added adequate details regarding the concentrations of the solutions used and the extracts. Since the plant extracts consist of a mixture of various bioactive compounds or phytochemicals with different polarities, we have to deal with a significant challenge in their separation and analysis. This complexity arises because plants produce a wide array of secondary metabolites, each serving different purposes such as defense mechanisms, attraction of pollinators, or responses to environmental stress or acting as hormones. The separation and identification of bioactive compounds from plant extracts involve techniques such as Thin-layer chromatography (TLC), column chromatography, flash chromatography, Sephadex chromatography, HPLC. These techniques are beyond the scope of this study. Please refer to Lines 300-302.

  1. Results and Discussion:

The results section presents a thorough analysis of the antibacterial and antioxidant efficacy of the Rosa damascena and Hypericum perforatum extracts against pathogenic strains. The data is well-organized and visually represented, aiding in a clear understanding of the findings. The discussion succinctly interprets the results and relates them to the existing literature. I suggest the authors consider expanding on the potential mechanisms of action underlying the observed antimicrobial and antioxidant activity to provide a more insightful discussion.

Response: Thank you for your comment. The specific mode of action could be the subject for a future article since no precise methodology was employed during this study. However, some thoughts based on the results from published studies, have been presented in our manuscript at the discussion part.

  1. Conclusion:

The conclusion aptly summarizes the key findings and their implications within the context of the study. However, I would encourage the authors to provide a brief statement on the limitations of the research and propose future directions to further investigate the potential clinical applications of the studied plant extracts.

Response: we have revised “limitations” and “conclusions” part

Thank you for your time and effort

The authors